# Challenges and Opportunities in the Catalytic Synthesis of Diphenolic Acid and Evaluation of Its Application Potential

**DOI:** 10.3390/molecules29010126

**Published:** 2023-12-24

**Authors:** Sara Fulignati, Nicola Di Fidio, Claudia Antonetti, Anna Maria Raspolli Galletti, Domenico Licursi

**Affiliations:** 1Department of Chemistry and Industrial Chemistry, University of Pisa, Via Giuseppe Moruzzi 13, 56124 Pisa, Italy; sara.fulignati@unipi.it (S.F.); nicola.difidio@unipi.it (N.D.F.); claudia.antonetti@unipi.it (C.A.); domenico.licursi@unipi.it (D.L.); 2Consorzio Interuniversitario Reattività Chimica e Catalisi (CIRCC), Via Celso Ulpiani 27, 70126 Bari, Italy

**Keywords:** diphenolic acid, levulinic acid, phenol derivatives, homogeneous catalysis, heterogeneous catalysis, epoxy resins, polycarbonates

## Abstract

Diphenolic acid, or 4,4-bis(4-hydroxyphenyl)pentanoic acid, represents one of the potentially most interesting bio-products obtainable from the levulinic acid supply-chain. It represents a valuable candidate for the replacement of bisphenol A, which is strongly questioned for its toxicological issues. Diphenolic acid synthesis involves the condensation reaction between phenol and levulinic acid and requires the presence of a Brønsted acid as a catalyst. In this review, the state of the art related to the catalytic issues of its synthesis have been critically discussed, with particular attention to the heterogeneous systems, the reference benchmark being represented by the homogeneous acids. The main opportunities in the field of heterogeneous catalysis are deeply discussed, as well as the bottlenecks to be overcome to facilitate diphenolic acid production on an industrial scale. The regioselectivity of the reaction is a critical point because only the *p*,*p*′-isomer is of industrial interest; thus, several strategies aiming at the improvement of the selectivity towards this isomer are considered. The future potential of adopting alkyl levulinates, instead of levulinic acid, as starting materials for the synthesis of new classes of biopolymers, such as new epoxy and phenolic resins and polycarbonates, is also briefly considered.

## 1. Introduction

Levulinic acid (LA) is a well-known platform chemical that is obtainable nowadays from cheap and waste cellulosic biomasses, thus allowing its industrial production on a larger scale [1,2]. The growth of LA production is allowing the progressive reduction of its manufacturing costs, thus favoring the research and development of new added-value bio-products derived from its supply-chain. At the state of the art, many technological improvements have been made by GF Biochemicals [3] to lower the LA production costs, aiming at a target of USD 1/kg, which is highly competitive in the market [4]. One of the most important added-value LA derivatives is 4,4-bis(4-hydroxyphenyl)pentanoic acid, also named diphenolic acid (DPA), for which the main physico-chemical properties are reported in Table 1.

Due to the structural similarities, DPA has been identified as a potential replacement for bisphenol A (BPA), the latter representing one of the most adopted monomers for the synthesis of epoxy resins and polycarbonates [5,6], which are widely exploited for the formulation of paints, cosmetics, surfactants, plasticizers, textile chemicals, etc. [7,8]. However, BPA is well-known for its dangerous toxicological properties, which mime the effects of estrogens and thus disrupt the endocrine system, even at very low concentrations (<1 ng L^−1^) [9,10]. Moreover, BPA use has been associated with the onset of health issues, such as cancer, obesity, and diabetes [11,12], and its use, especially in foodstuffs, is strictly regulated by European Food Safety Authority (EFSA) legislation [13]. In this context, DPA represents an excellent substitute for BPA, and the progressive lowering of the LA production costs makes further R&D on this topic particularly valuable. Moreover, DPA can be envisaged as a completely bio-based molecule if the phenol production from renewable lignin sources, mainly through pyrolysis or reductive depolymerization, will be implemented on an industrial scale [14,15]. Remarkably, such pathways could be even better tailored to give substituted phenols (catechol, resorcinol, cresol, guaiacol, etc.), thus further broadening the application opportunities of the final specialty biomaterials. Whilst BPA is produced through a condensation reaction between acetone and two moles of phenol, in the DPA synthesis, acetone is replaced by LA, and both reactions require the presence of a proper Brønsted catalyst. A comparison between the reactions involved in the BPA and DPA syntheses is shown in Figure 1.

According to Rahaman et al. [16], the reaction mechanism of DPA synthesis (Figure 2) involves the preliminary formation of the mono-phenolic acid (MPA) intermediate via activation of the LA ketone group, followed by the subsequent nucleophilic attack of the phenol and finally through the acid-catalyzed loss of a water molecule. The obtained carbocation intermediate (MPA) undergoes the nucleophilic attack of another phenol molecule to obtain the target DPA.

Just as for BPA, the synthesis of DPA is traditionally carried out in the presence of strong mineral acids, such as concentrated HCl or H_2_SO_4_, which are generally considered benchmarks for the development of new investigations. Remarkably, an isolated yield of about 60 mol% (evaluated with respect to the starting LA) was claimed in Bader’s patent [17], typically employing an LA/phenol molar ratio of 1/2 and working at 25 °C for 20 h with concentrated H_2_SO_4_ (about 80 wt%). Such molar yields were also claimed employing the same LA/phenol molar ratio and a lower concentration of HCl (about 25 wt%) but working at a higher reaction temperature (about 90 °C) for 6 h. The use of HCl is generally preferred over H_2_SO_4_ due to the simpler work-up operations, which are required for its recovery. More recently, the use of concentrated mineral acids (37 wt% HCl or 96 wt% H_2_SO_4_), excess of phenol (LA/phenol molar ratio within the range 1/2–1/4), mild reaction temperatures (up to 60 °C), and long reaction times (up to 48 h) have generally been proposed, thus achieving a maximum DPA yield of about 70 mol% [18,19,20,21,22]. Remarkably, the use of a thiol (namely, methyl mercaptan) as a reaction co-catalyst led to significant improvement of the DPA yield up to a maximum of about 93 mol%, working at 55 °C for 16 h, in the presence of 37 wt% HCl as the main catalyst, and adopting the LA/phenol molar ratio of 1/2 [23]. The chemistry involved in such reactions was not explained in detail by the authors, and claims of such patents were mainly based on experimental evidence. Moreover, the reaction is mostly performed under solvent-free conditions, e.g., without any addition of external water and/or other solvents, thus exploiting the use of an excess of phenol. To specifically assess issues related to the selectivity of the DPA synthesis, it is necessary to consider more recent works, including, in some cases, the chemistry of BPA as the only reference, which admits the formation of different degradation products, mainly including triphenols and chromans [24]. In any case, the identification of the corresponding by-products in the DPA synthesis has not been specifically discussed in the literature, and scarce information is available on a few reaction by-products. In this context, according to the reaction mechanism proposed in Figure 2, the nucleophilic attack of the second molecule of phenol may lead to the formation of the *o*,*p*′-DPA isomer (Figure 3), and this isomer is considered an unwanted reaction by-product because it has no industrial interest and cannot be recycled within the process. In such condensation reactions involving unsubstituted phenol, the formation of the *o*,*o*′-isomer is almost negligible for steric hindrance, whilst those of the *p*,*p*′- and *o*,*p*′- ones are both thoroughly allowed [25]. The presence of the *o*,*p*′-DPA isomer significantly complicates the purification procedures of the *p*,*p*′-DPA, which represents the only industrially useful product. On this basis, the *p*,*p*′/*o*,*p*′ molar ratio represents a key parameter for the optimization of the DPA synthesis process due to the difficult separation of the two isomers through crystallization. For the BPA synthesis, an isomerization reaction was planned at the industrial scale to advantageously convert the *o*,*p*′-BPA isomer to the *p*,*p*′-BPA one [26], but this isomerization is extremely difficult in the case of DPA isomers; thus, this crucial aspect requires the adoption of more appropriate strategies already during the synthesis step.

The *p*,*p*′-DPA/*o*,*p*′-DPA molar ratio also has important effects on the properties of the final polymers, such as the color stability, crystallinity, and intermolecular attractive forces between the polymer chains. Thus, the *p*,*p*′-DPA/*o*,*p*′-DPA molar ratio should be as high as possible to improve these properties [27]. Remarkably, the available data based on Bader’s patents, such as those of reference in this discussion [17], seem to confirm the exclusive formation of the *p*,*p*′-DPA isomer, but these data must be much more carefully considered due to the instrumental limitations at that time. In fact, the most recent literature admits the formation of the *o*,*p*′-DPA isomer starting from phenol [16], and, in this regard, different approaches have been proposed to improve the selectivity towards the *p*,*p*′-isomer. Remarkably, 2,6-xylenols, such as 2,6-dimethylphenol, were proposed as a phenolic source for the acid-catalyzed condensation with LA to the corresponding bis-xylenol product [28], thus completely addressing the reaction selectivity towards the corresponding *p*,*p*′-isomer (that, is 4,4-bis(3,5-dimethyl-4-hydroxyphenyl)pentanoic acid). Even more advantageously, natural phenols deriving from lignin sources, such as catechol and resorcinol, could be smartly exploited as valuable replacements for unsubstituted phenol [29]. In this case, in addition to the para-orientating effect, the presence of an additional hydroxyl group in the appropriate position of the phenolic structure is advantageous because it adds another reactive group to the DPA derivative, which could be better exploited to produce even more added-value products, such as glycidyl ethers [29].

Regarding the catalytic issues, although mineral acids are the traditional ones, their use has some well-known and important drawbacks, including corrosivity, difficult separation, and expensive treatment of the process’s wastewater. These general issues are stimulating the development of new catalysts to perform DPA synthesis with the intention of improving the activity and the selectivity towards the *p*,*p*′-DPA isomer and, above all, overcoming the corrosion and separation problems. In this context, sulfonated acid catalysts [19,27,30,31], heteropolyacids [16,18,20,30,32], zeolites and oxides [18,27], as well as ionic liquids [21,22,33] have been proposed for DPA synthesis in the search for further improvement of the selectivity to the *p*,*p*′-DPA isomer. In this context, the use of a thiolic derivative as a co-catalyst was proposed by Van De Vyver et al. [30,31] to further improve the selectivity to the *p*,*p*′-DPA isomer. According to the reaction mechanism proposed by the authors for explaining the behavior of sulfonic-acid-functionalized catalysts, the thiol additive acts as a promoter by improving the reactivity of the carbonyl group of LA, thus making it more prone to react with phenol. Therefore, the thiol would react with the carbonyl of the LA to give an intermediate that is more electrophilic towards the reaction with phenol (Figure 2). The authors found that thiols with small substituents, such as ethanethiol, were particularly effective towards the *p*,*p*′-DPA formation, preferentially working at low LA conversion and therefore under kinetic control [31]. On the other hand, at higher LA conversions, the formation of the *o*,*p*′-DPA isomer became relevant according to a regime of thermodynamic control.

Starting from the above milestones, the state of the art related to the production of DPA from LA is critically discussed in this review by focusing on the development of new acid catalysts, with particular attention to the heterogeneous systems and considering the homogeneous ones as the benchmark. Subsequently, the main available applications of *p*,*p*′-DPA are summarized and discussed while highlighting the bottlenecks to be solved for further improving the research on this topic.

## 2. DPA Synthesis

### 2.1. DPA Synthesis with Homogeneous Catalysts

The condensation reaction between LA and phenol to give DPA has been traditionally carried out in the presence of strong Brønsted mineral acids, mainly HCl and H_2_SO_4_. The most interesting results reported in the literature with these mineral acids and other homogeneous systems, mainly referring to sulfonic acids, are reported in Table 2.

Very different reaction conditions in terms of the LA/phenol molar ratio, LA/catalyst weight ratio, reaction temperature, and time have been proposed when working with mineral acids and, in some cases, adding thiols to improve the selectivity towards the desired *p*,*p*′-DPA isomer [21]. However, H_2_SO_4_ generally led to higher yields and *p*,*p*′-DPA/*o*,*p*′-DPA molar ratios compared to HCl, regardless of the adopted reaction conditions (entries 1–7, Table 2). Thus, H_2_SO_4_ results are more promising for this application. On this basis, it is possible to suppose that the sulfonic group plays a key role in the reaction. For this reason, many other sulfonic acids have been proposed, including *p*-toluenesulfonic acid (entries 8 and 9, Table 2), sulfamic acid (entry 10, Table 2), methanesulfonic acid (entries 11 and 12, Table 2), triflic acid (entry 13, Table 2), and 3-mercaptopropyl sulfonic acid (entry 14, Table 2). In this context, Shen et al. [21] compared the catalytic performances of *p*-toluenesulfonic acid and sulfamic acid with those of HCl (entries 8, 10, and 3, Table 2) working under the same reaction conditions, including the amount of catalyst and ethanethiol (as an additive). The authors found that HCl led to the highest LA conversion (entry 3, Table 2, 65 mol%) but also to the lowest *p*,*p*′-DPA/*o*,*p*′-DPA molar ratio (2.0), whilst the sulfamic acid resulted in the most promising catalyst (entry 10, Table 2), showing comparable conversion but with a much higher selectivity towards the *p*,*p*′-DPA isomer (*p*,*p*′-DPA/*o*,*p*′-DPA molar ratio of 7.4) and the yield of 51 mol%. Mthembu et al. [33] optimized the Response Surface Methodology (Box–Behnken) for the condensation reaction between commercial LA and phenol by employing methanesulfonic acid as the homogeneous catalyst. The effects of the reaction time, temperature, and acid loading were investigated while keeping constant the LA/phenol molar ratio (3.7). The highest overall DPA yield of 66 mol%, corresponding to a yield of the *p*,*p*′-DPA isomer of 37 mol%, was achieved by working for 6 h at 75 °C with a LA/catalyst ratio of 0.9 wt/wt (entry 11, Table 2). The same authors also tested H_2_SO_4_ (entry 5, Table 2) and *p*-toluenesulfonic acid (entry 9, Table 2), in both cases achieving higher overall DPA yields than that reported for the methanesulfonic acid while working under the same reaction conditions. However, the authors did not report the *p*,*p*′-DPA/*o*,*p*′-DPA molar ratio reached with H_2_SO_4_ and *p*-toluenesulfonic acid; thus, further specific information on the *p*,*p*′-DPA yield is not available. The increment of the amount of phenol up to an LA/phenol ratio of 1/9.2 mol/mol was effective for improving the overall DPA yield from 66 to 86 mol% (entry 12, Table 2) but, in this case as well, no further information was given on the *p*,*p*′-DPA isomer yield. Moreover, the authors synthesized LA from depithed sugarcane bagasse as a real waste biomass to obtain an LA-rich liquor, working for 7 h at 100 °C in the presence of [EMim][HSO_4_] as both solvent and catalyst, thus achieving an LA yield of 55 mol%. The crude LA-rich mixture further reacted with phenol under the same reaction conditions (75 °C, 6 h, and 1/3.7 as the LA/phenol molar ratio), thus achieving a total DPA yield of 65 mol% and proving that even crude LA derived from waste biomass can be directly used for DPA production. Liu et al. compared the catalytic activity of H_2_SO_4_ (entry 6, Table 2), triflic acid (entry 13, Table 2), and 3-mercaptopropyl sulfonic acid (entry 14, Table 2) by keeping constant the reaction conditions (60 °C for 48 h and 1/4 as the LA/phenol molar ratio), reaching the highest *p*,*p*′-DPA yield and *p*,*p*′-DPA/*o*,*p*′-DPA molar ratio by working with 3-mercaptopropyl sulfonic acid (entry 14, Table 2) [22]. Considering the promoting effect of the thiol groups, the mercaptoacetic acid was employed as an additive for the H_2_SO_4_-catalyzed reaction (entry 7, Table 2), which only modestly improved the *p*,*p*′-DPA yield from 61 to 68 mol%, but a reduction of the *p*,*p*′-DPA/*o*,*p*′-DPA molar ratio was also ascertained. According to the available literature [31], the formation of the *o*,*p*′-DPA isomer becomes relevant at high LA conversion under a regime of thermodynamic control, whereas the use of thiol additives for improving the *p*,*p*′-DPA production can be advantageous at low LA conversions, therefore working under kinetic control.

Isolation of DPA from a crude reaction mixture is a challenging topic, and only few examples are available in the literature, which mainly apply to DPA synthesized in the presence of a mineral acid (H_2_SO_4_ and HCl) as the catalyst. According to Bader [17], the isolation of DPA from such a reaction environment can be realized by diluting the crude reaction mixture with water and, next, extraction with ethyl acetate. The extract (containing phenol and DPA) can be further extracted with an aqueous solution of sodium bicarbonate to selectively deprotonate the DPA, which becomes soluble in the water phase. The latter is acidified and extracted with ether, and the corresponding extract is stripped in vacuo to yield the isolated DPA. Depending on the required purity degree, crude DPA can be further crystallized, generally with organic solvents, such as aromatic hydrocarbons (toluene), but also with water or ethanol.

### 2.2. DPA Synthesis with Heterogeneous Sulfonated Systems

Starting from the previous discussion, the use of heterogeneous catalysts that include sulfonic groups has represented the smartest choice to be immediately developed. In this context, many authors have proposed the use of heterogeneous systems, including the sulfonic acid group, to perform the DPA synthesis. The main available results based on such commercial and ad hoc synthesized catalysts are reported in Table 3.

Among the commercial, heterogeneous, sulfonic-acid-based systems, Amberlyst-15 is the preferred choice according to entries 15–17 of Table 3, with the highest *p*,*p*′-DPA yield (55 mol%) and *p*,*p*′-DPA/*o*,*p*′-DPA molar ratio (15.8), which were ascertained while working at 60 °C for 24 h and employing an LA/phenol molar ratio of 1/4 and LA/catalyst weight ratio of 4.4. Remarkably, comparing the results obtained with the same LA/phenol ratio (1/4, according to entries 15 and 16), it is evident that a lower reaction temperature should be preferred while compensating with a greater amount of the catalyst to obtain a similar conversion of LA but significantly better selectivity control towards the *p*,*p*′-DPA formation. Van De Vyver et al. [30] compared the catalytic activity of the Amberlyst-15 with that of Nafion NR50 (entry 18, Table 3), highlighting that the latter led to a higher *p*,*p*′-DPA/*o*,*p*′-DPA molar ratio while achieving similar LA conversion under the same reaction conditions. Moreover, the authors synthesized a new class of sulfonated hyperbranched poly(arylene oxindole)s (SHPAOs), which were tested for DPA synthesis. The macromolecular structure of these hyperbranched polymers positively acts on the increase in the functional group density, which is a desirable aspect for developing catalytic applications. When such catalysts were employed for the DPA synthesis, better performances than those of Amberlyst-15 and Nafion NR50 were achieved, mainly in terms of higher conversion and selectivity to the *p*,*p*′-DPA isomer (entry 19, Table 3). In addition to the desired *p*,*p*′-DPA product, the authors reported the formation of the *o*,*p*′-DPA isomer, as well as oligomeric phenol derivatives and LA dimers [34]. Such condensation reactions are typical of the acid catalysis approach, such as those occurring during BPA synthesis [35]. These side-reactions involving LA, which represents the limiting reagent of DPA synthesis, must be avoided, as they negatively impact *p*,*p*′-DPA selectivity. To selectively favor *p*,*p*′-DPA production, Van De Vyver et al. [30] proposed the use of thiol as the additive, thus showing an improvement in LA conversion and selectivity towards the *p*,*p*′-DPA isomer (entries 20–25, Table 3). By adopting thiols with different chain lengths, the authors found that the phenol condensation was the rate-determining step, and the condensation rate decreased in the following order: ethanthiol > 1-propanethiol ≈ benzylthiol > 1-butanthiol > 2-propanethiol > 2-methyl-2-propanethiol. This suggests that the reaction rate is significantly affected by the thiol steric hindrance. These results agree with the work of Margelefsky et al. [36], who proposed that the formation of the electrophilic sulfonium intermediate accelerates the condensation rate towards the target product, thus preferentially shifting the regioselectivity towards the *p*,*p*′-DPA isomer thanks to the introduction of minor steric hindrance of the thiol side chain. In this context, the increase in the thiol steric hindrance significantly worsens the catalytic activity and *p*,*p*′-DPA selectivity, making necessary the employment of an accessible thiol group [30]. Moreover, it is well-known that the *p*,*p*′-DPA can isomerize to the *o*,*p*′-DPA isomer in the presence of acid catalysts; thus, Van De Vyver et al. investigated in greater depth the effect of thiol additives on this isomerization reaction [31]. They found that thiols scarcely affected the isomerization step and the second condensation with phenol to give DPA, whilst such additives play a key role in determining the regioselectivity of the first condensation step of LA with phenol to give MPA. An appropriate balance between the acidity of the catalyst and the steric hindrance and the amount of the thiol additive represents the right solution to achieve high *p*,*p*′-DPA yields. In particular, the choice of a low SO_3_H/SH ratio is effective in favoring the reaction rate and increasing the *p*,*p*′-DPA/*o*,*p*′-DPA molar ratio (compare entry 23 with entry 26, Table 3). Based on such promising results, the authors also investigated the post-synthetic modification of SHPAOs by incorporating aminothiols, such as 2-mercaptoethylamine (SHPAOs-MEA) and 4-(2-thioethtyl)-pyridine (SHPAOs-TEP) (entries 27 and 28, Table 3). Both of these catalysts led to a significant improvement in the LA conversion and regioselectivity control to the *p*,*p*′-DPA when compared with the absence of the thiol additive (entry 19, Table 3). More recently, Zhu et al. [19] prepared magnetic catalysts to be employed in the DPA synthesis from LA. In this context, an N-doped carbon nanotube modified with -SO_3_H groups encapsulating Fe, Ni, or Co nanoparticles (Fe@NC-SO_3_H, Ni@NC-SO_3_H, Co@NC-SO_3_H) were synthesized. All of the investigated heterogeneous catalysts gave the best performances in terms of higher LA conversions, *p*,*p*′-DPA/*o*,*p*′-DPA molar ratios, and *p*,*p*′-DPA yields (entries 29–31, Table 3) compared to those achieved with H_2_SO_4_ under the same reaction conditions (entry 4, Table 2). Remarkably, Co@NC-SO_3_H was the most promising catalyst, leading to the highest *p*,*p*′-DPA yield of 63 mol% (entry 31, Table 3). The authors attributed these valuable catalytic performances to the regular microstructure and larger specific surface area of Co@NC-SO_3_H, which resulted in a more uniform distribution and easier accessibility of the SO_3_H groups on the catalyst’s surface. At the end of the reaction, the catalyst was magnetically separated and employed in recycling tests, showing good performances after four cycles. The authors attributed the high stability of this catalyst to its encapsulated structure, which protects the internal metal nanoparticles from unwanted degradation processes, such as leaching. The stability of the recovered nanotubes was experimentally confirmed by verifying the absence of Co in the reaction medium, the uniform distribution of -SO_3_H groups on the catalyst’s surface, and the presence of encapsulated Co nanoparticles. In addition, Zhu et al. [19] studied the role of thiol in the improvement of *p*,*p*′-DPA production by adopting the same Co@NC-SO_3_H system. Again, a small thiol (namely, mercaptoacetic acid) was effective in improving both LA conversion and the *p*,*p*′-DPA/*o*,*p*′-DPA molar ratio (entry 32, Table 3). The adoption of longer reaction times (from 24 to 48 h) improved the LA conversion and the *p*,*p*′-DPA yield (91 mol%), keeping almost constant the *p*,*p*′-DPA/*o*,*p*′-DPA molar ratio (entry 33, Table 3). The authors also investigated the catalytic performances of mercaptoacetic acid and Co@NC alone to exclude any relevant catalytic activity due to their individual contributions. In both cases, the LA conversion was lower than 5 mol% and no DPA was detected, thus proving that mercaptoacetic acid had a too-low acidity and that the -SO_3_H groups were the effective active sites of the catalyst. An overall evaluation of the data related to sulfonated catalysts highlights good catalytic performances for the Amberlyst-15 sulfonic resin and, even better, excellent performances for the synthesized Co@NC catalytic system, which is easily recoverable and thermally stable.

### 2.3. DPA Synthesis with Heteropolyacids

The heteropolyacids are another class of acid catalysts that have been employed in the synthesis of DPA. They have a higher acid strength than traditional solid acids (oxides and zeolites) and are less corrosive than mineral ones [16]. On this basis, several types of heteropolyacids have been synthesized and studied for DPA synthesis (Table 4) while working in the absence of thiol additives.

Rahaman et al. [16] prepared different Keggin-type heteropolyacids ([XM_12_O_40_]^n−^), including phosphotungstic acid (H_3_PW_12_O_4_), silicotungstic acid (H_4_SiW_12_O_40_), and silicomolybdic acid (H_4_SiMo_12_O_4_), and compared their catalytic performances under the same reaction conditions (entry 34–36, Table 4). The authors proved that all of the tested catalysts were active in LA conversion, which was always higher than 55 mol%, but only H_3_PW_12_O_4_ gave a good DPA yield (46 mol%) while at the same time showing higher selectivity to the *p*,*p*′-DPA. Because of the higher productivity obtained with H_3_PW_12_O_4_, it was employed for further optimization of the reaction. Regarding the catalyst amount, the authors found that its increased sped up LA conversion, but it did not affect the selectivity towards DPA. On the other hand, the temperature increase was effective for improving the LA conversion and the *p*,*p*′-DPA/*o*,*p*′-DPA molar ratio, thus proving that the regioselectivity control of this reaction greatly depends on the temperature. The highest *p*,*p*′-DPA/*o*,*p*′-DPA molar ratio (28.3) was reached at 140 °C, together with LA conversion of 87 mol% and a *p*,*p*′-DPA yield of 82 mol% (entry 37, Table 4). At the state of the art, H_3_PW_12_O_4_ is the most studied heteropolyacid for the synthesis of DPA, and it was reported as the catalyst of reference in the works of Li et al. [20], Van De Vyver et al. [30], and Yu et al. [18]. However, lower LA conversions and *p*,*p*′-DPA yields were reported in all of these studies (entries 38–40, Table 4) when compared with the results of Rahaman et al. (entry 37, Table 4) [16]. Interestingly, Yu et al. [18] also employed Wells–Dawson phosphotungstic acid (H_6_P_2_W_18_O_62_), claiming higher LA conversion, *p*,*p*′-DPA/*o*,*p*′-DPA molar ratio, and *p*,*p*′-DPA yield (entry 41, Table 4) with respect to the Keggin phosphotungstic acid (H_3_PW_12_O_4_). Some bottlenecks still limit the employment of heteropolyacids, including small specific surface area (1–5 m^2^/g) and good solubilities in polar solvents, which limit their recovery/reuse. To overcome these issues, some smart strategies have been proposed, such as their immobilization on solid supports and the exchange of protons with large alkali cations, to form insoluble salts. The first approach was proposed by Guo et al. [32], who prepared some mesostructured silica-supported H_3_PW_12_O_4_ catalysts with different heteropolyacids loadings. This goal was achieved through the contemporary hydrolysis and condensation of tetraethoxysilane (TEOS) with H_3_PW_12_O_4_ in the presence of a template surfactant (Pluronic P123), followed by the hydrothermal treatment and template removal through calcination or extraction with boiling ethanol. This one-pot synthesis is smarter than the traditional post-synthesis grafting method, because the latter results in poor control of the heteropolyacid loadings and not negligible leaching phenomena. Moreover, the employment of a non-ionic surfactant (Pluronic P123) is effective for weakening the interactions between the template and the inorganic walls; thus, it can be easily removed (through solvent extraction or calcination at low temperatures) without damaging the catalyst structure. The authors investigated the effect of the H_3_PW_12_O_4_ loading, finding that its increment was effective for improving the LA conversion and *p*,*p*′-DPA yield, but a slight decrease in the *p*,*p*′-DPA/*o*,*p*′-DPA molar ratio was also observed (entries 42–45, Table 4). Moreover, the catalytic activities of the catalysts having the same H_3_PW_12_O_4_ loading but resulting from different template removal were compared, demonstrating that the extracted catalyst (H_3_PW_12_O_40_/SiO_2_-E-7.5, entry 43, Table 4) led to higher LA conversion and *p*,*p*′-DPA/*o*,*p*′-DPA molar ratio than the calcined one (H_3_PW_12_O_40_/SiO_2_-C-7.5, entry 46, Table 4). The authors ascribed this different catalytic behavior to the different textural properties, and, in particular, to the narrower pore sizes of H_3_PW_12_O_40_/SiO_2_-C-7.5 (6.0 nm) with respect to H_3_PW_12_O_40_/SiO_2_-E-7.5 (7.2 nm). Given the large dimensions of the DPA molecule, a catalyst with a large pore size is more suitable for allowing the accessibility of the reactants on the acid sites, as well as the next desorption of the produced DPA from its surface. On the other hand, the narrow pores may cause an increment of the internal mass transfer limitation and, consequently, slower substrate conversion. Moreover, the calcination of the catalyst at 420 °C caused the loss of some acid sites, leading to a lower activity of H_3_PW_12_O_40_/SiO_2_-C-7.5 with respect to H_3_PW_12_O_40_/SiO_2_-E-7.5. Finally, the authors investigated the recyclability of the prepared catalysts. At the end of the reaction, the catalyst was separated through filtration and then washed with water and ethanol and calcined at 120 °C under vacuum for 1 h. This spent catalyst was employed within two following reactions, partially losing the catalytic performances after the first recycle run and completely losing it after the second one. This evident worsening aspect was not due to the leaching of H_3_PW_12_O_40_ (its absence in the reaction medium was verified through ICP-AES analysis), but rather the strong adsorption of DPA on the catalyst surface. This hypothesis was confirmed by restoring the initial activity through calcination of the spent catalyst at higher temperatures. The same authors deeply investigated the synthesis of these H_3_PW_12_O_40_-silica materials and, properly adjusting the composition ratios of the precursors, temperature, and acidity of the solution, they prepared well-defined ordered mesoporous catalysts with a 2D or 3D hexagonal structure and tested them in the synthesis of DPA [20]. First, they studied the influence of the LA/phenol molar ratio; in principle, an excess of phenol was employed, but molar ratios that are too low could limit DPA formation due to the dilution effect of phenol towards LA. Once having identified the best LA/phenol molar ratio (fixed at 1/7), the authors studied the influence of the H_3_PW_12_O_40_ loading supported on the 3D hexagonal structure (entries 47–53, Table 4). The highest *p*,*p*′-DPA yield of 14 mol% was achieved with the 11.1 wt% loading of H_3_PW_12_O_40_, highlighting that lower loading gave insufficient active sites and that the reaction was particularly slow. On the other hand, higher loadings led to the aggregation of heteropolyacid units at the catalyst surface, thus reducing the available surface of the active sites. Moreover, considering the catalysts with similar H_3_PW_12_O_40_ loading obtained through the same template removal method, the 3D hexagonal silica material led to higher catalytic activity than the 2D hexagonal one (compare entries 48 and 54, Table 4). In fact, the 3D morphology allows for more efficient transport of the reactant molecules in many more directions through an easier diffusion than the 2D morphology. Lastly, in the same work, the authors confirmed that the template removal through the extraction route should be preferred, as it allows for a higher structural ordering and catalytic activity (compare entries 54 and 55, Table 4). The recyclability of the best-performing catalyst (H_3_PW_12_O_40_/SiO_2_-3D_hex_-C-11.1) was investigated and, as in the previous case, a bulk loss of activity after three runs was ascertained, which was mainly attributed to the strong adsorption of the DPA molecules on the catalyst surface. However, after calcination at 420 °C for 1 h, the pristine catalytic activity was almost completely restored. In developing a different approach, protons of heteropolyacids are often exchanged with large alkali cations, thus obtaining the corresponding insoluble salts in the polar reaction medium and improving the catalyst recovery. In this context, Yu et al. [18] partly substituted the protons of Keggin (Cs_x_H_3−x_PW_12_O_40_ with x = 1.0–2.5) and Wells–Dawson (Cs_x_H_6−x_P_2_W_18_O_62_ with x = 1.5–4.5) heteropolyacids with cesium, and they found that the latter led to higher *p*,*p*′-DPA yields and *p*,*p*′-DPA/*o*,*p*′-DPA molar ratios than the Keggin-type catalysts when tested under the same reaction conditions (entries 56 and 57, Table 4). This different behavior depended on the different reaction mechanisms of Wells–Dawson and Keggin-type catalysts. In fact, the adsorption ability of Wells–Dawson heteropolyacids strongly depended on the content of cesium, which gradually decreased by increasing the cesium content, whilst the adsorption ability of Keggin heteropolyacids was related to their specific surface area. These properties suggested that with the Wells–Dawson catalysts, the reaction proceeded in the pseudo-liquid phase; thus, the polar molecules as LA can be rapidly adsorbed into the catalyst and react there. This pseudo-liquid phase mechanism often allows for reaching high catalytic activity and promising selectivity. On the other hand, with the Keggin heteropolyacids, the reaction proceeded according to a surface-type mechanism, where the adsorption–desorption step was generally slow, thus justifying the more interesting results achieved with the Wells–Dawson catalysts. The authors selected the most active catalysts for the Wells–Dawson and Keggin-type heteropolyacids (namely, Cs_1.5_H_4.5_P_2_W_18_O_62_ and Cs_2.5_H_0.5_PW_12_O_40_, respectively) and studied the influence of several reaction parameters. They found that the increment of phenol improved the DPA yield without any appreciable change in selectivity to the *p*,*p*′-DPA isomer. On this basis, the optimal LA/phenol molar ratio was fixed to 1/9 and 1/4 for the Cs_1.5_H_4.5_P_2_W_18_O_62_ and Cs_2.5_H_0.5_PW_12_O_40_, respectively. Analogously, the increment in the catalyst amount promoted the DPA yield, together with a negligible effect on the reaction selectivity. On the contrary, the temperature affected not only the DPA yield but also the *p*,*p*′-DPA selectivity. Moving from 80 to 150 °C, the *p*,*p*′-DPA selectivity increased from 70 to 85 mol% for the Keggin-type heteropolyacid and from 76 to 89 mol% for the Wells–Dawson one. Moreover, the authors proved the presence of external diffusion limitations with both catalysts by changing the stirring speed from 800 to 1200 rpm, with the latter further improving the DPA yield. Once having optimized the reaction conditions, the highest *p*,*p*′-DPA yields obtained with Wells–Dawson and Keggin-type heteropolyacids were 62 and 37 mol%, respectively (entries 58 and 59, Table 4). The recyclability of these heteropolyacids was investigated, and, for this purpose, the recovered catalysts were washed with ethanol and calcined at 100 °C for 1 h. The catalytic performances did not significantly change during three recycle tests, and the ICP-AES analysis of the reaction mixtures confirmed the absence of leaching phenomena, thus demonstrating the catalyst’s stability. In conclusion, considering the available data on heteropolyacids, the best catalytic performances are achieved with cesium-based systems combining high yields/selectivity to the *p*,*p*′-DPA, with an excellent advantage for its recovery/reuse.

### 2.4. DPA Synthesis with Other Heterogeneous Systems: Zeolites and Modified Metal Silicas

Acid zeolites and metal oxides are often exploited in the field of acid catalysis due to their mild acidity, which is effective for reducing corrosion issues, as well as being more easily recoverable from the reaction medium and, in many cases, easily recyclable. Their catalytic performances strongly depend on the nature, number, and distribution of acid sites, differentiating them from Brønsted and Lewis ones. Such heterogeneous catalysts have also been employed for DPA synthesis (Table 5) working in the absence of thiol additives.

Morales et al. [27] compared the catalytic activity of different modified metal oxides (propylsulfonic-acid mesostructured silica, arenesulfonic-acid functionalized mesostructured silica, and Nafion supported mesostructured silica, entries 60–62, Table 5), zeolites (n-ZSM-5, H-USY and H-Beta, entries 63–65, Table 5), and sulfonic-acid-based resins (Amberlyst-15, entry 15, Table 3) while keeping constant the reaction conditions. Amberlyst-15 was found to be the most active catalyst, leading to the highest LA conversion (64 mol%) but to a very low *p*,*p*′-DPA yield (6 mol%), thus highlighting a very poor selectivity. Similarly, the modified SBA-15 silicas showed modest LA conversions and very low DPA yields. These results can be attributed to the presence of strong Brønsted acid sites that promoted undesired pathways, thus causing the LA conversion to by-products. However, the authors did not provide further insight into such by-products. On the other hand, among the tested zeolites, the H-Beta allowed for the achievement of the highest LA conversion of 44 mol%, a *p*,*p*′-DPA yield of 33 mol%, and a *p*,*p*′-DPA/*o*,*p*′-DPA molar ratio of 99 when performing the reaction at 120 °C for 24 h (entry 65, Table 5), thus exploiting the appropriate combination of pore size, structure, type, and strength of acid sites. Moreover, a marked difference between the selectivity of the sulfonic catalysts and zeolites was ascertained: the former preferentially led to the formation of the *o*,*p*′-DPA, whereas the latter led to the desired *p*,*p*′-DPA. Because the best catalytic performances were achieved with the H-Beta zeolite, the authors focused on the investigation of the influence of the Si/Al ratio for this type of system by increasing it from 12.5 to 180 (entries 65–68, Table 5). The H-Beta 75 was the most active catalyst, thus leading to the highest LA conversion and DPA yield, but the selectivity, equal to 78 mol%, was lower than that achieved with the H-Beta 19 (84 mol%). The authors assigned the best catalytic performances of the H-Beta 19 due to the right Brønsted/Lewis acid sites ratio, which was even more thoroughly evaluated by considering the strong acid sites (strong Brønsted/strong Lewis acid sites ratio equal to about 2.0). After the optimization of the reaction using the Response Surface Methodology, the authors claimed that all of the investigated independent variables (namely, temperature, H-Beta 19 loading, and LA/phenol molar ratio) positively affected the LA conversion and overall DPA yield. The highest overall DPA yield of 70 mol%, with the corresponding LA conversion of 77 mol%, was achieved after 72 h at 140 °C by employing the LA/phenol molar ratio of 1/6. It is noteworthy that the excellent *p*,*p*′-DPA/*o*,*p*′-DPA molar ratio of 99 was kept under these reaction conditions, corresponding to a *p*,*p*′-DPA yield of 69 mol% (entry 69, Table 5). The recyclability of the H-Beta 19 catalyst was investigated in three subsequent runs after washing with acetone and drying at room temperature overnight. The recycled catalyst showed a decrease in DPA yield up to 28 mol% after the third run, with a negligible change in the LA conversion, thus highlighting a loss of selectivity. The authors proved the presence of organic compounds on the catalyst surface, which are responsible for the deactivation issues. However, the pristine activity was almost totally restored through calcination in air at 550 °C for 5 h.

On the basis of the above data, the use of zeolites for this condensation reaction is promising, considering that the zeolite pore size and structure must be adequate for the molecules of interest, as well as the type and strength of the acidity. In this regard, a moderate strength of acid sites should be preferred for *p*,*p*′-DPA synthesis in order to avoid undesirable side-reactions, which typically occur in the presence of strong Brønsted acid catalysts. For this purpose, the Si/Al ratio represents the key parameter affecting the acid properties of these catalysts, determining not only the amount and concentration of acid sites but also their nature (Brønsted and Lewis) and strength. Therefore, Beta zeolite with a moderate aluminum content (H-Beta 19, Si/Al = 23) represents the best catalyst to perform the solvent-free condensation between LA and phenol owing to the shape selectivity conferred by its structure and the adequate balance of acidity (Al content and speciation).

### 2.5. DPA Synthesis with Other Catalysts: Ionic Liquids

Ionic liquids are well-known, robust, and designable organic salts typically composed of large cations and small anions that can find applications as solvents and catalysts for carrying out the syntheses of several bio-based compounds [37]. In the field of DPA synthesis, acid ionic liquids have been proposed for the condensation reaction between LA and phenol. The most significant results available for such catalysts are summarized in Table 6.

Shen et al. [21] tested different ionic liquids, again proposing the use of ethanethiol as the additive for further improving the catalytic performances. Remarkably, it was found that [BSMim]CF_3_SO_3_ was the most active and selective towards the *p*,*p*′-DPA isomer, reaching the maximum LA conversion of 81 mol% and the *p*,*p*′-DPA yield of 79 mol% (entry 70, Table 6), a promising result attributed to the in situ formation of HF, which is mainly responsible for the acid catalysis. [BSMim]OAc and [BSMim]HSO_4_ were also active, but the former led to only moderate results because of its lower acidity, thus demonstrating the key role of the anion (entries 71 and 72, Table 6). However, the cation also strongly influenced the catalytic activity of the ionic liquids, as shown by [BSMim]HSO_4_ and [BPy]HSO_4_ systems (entries 72 and 73, Table 6), with the former leading to higher LA conversion and *p*,*p*′-DPA yield due to the presence of the sulphonic acid in the [BSMim]HSO_4_ cation. The importance of -SO_3_H groups in the cation/anion of the ionic liquid is also highlighted by the lower LA conversion, *p*,*p*′-DPA yield, and *p*,*p*′-DPA/*o*,*p*′-DPA molar ratio achieved with [AMim]Br and [BMim]Cl (entries 74 and 75, Table 6). Moreover, the authors investigated the effect of the additive (ethanethiol) for the improvement of the catalysis with [BSMim]CF_3_SO_3_, which was identified as the most active catalyst (entry 70, Table 6). For the test carried out in the absence of the additive (entry 76, Table 6), a significant worsening of the *p*,*p*′-DPA/*o*,*p*′-DPA molar ratio was ascertained, thus confirming that the regioselectivity of the reaction depends on the synergy between the type of ionic liquid and additive. However, although [BSMim]CF_3_SO_3_ was identified as the best catalyst, its synthesis is difficult and expensive; thus, the optimization of the DPA synthesis was developed employing [BSMim]HSO_4_ as the catalyst, which allowed for achieving similar promising results (entry 72, Table 6). Temperature, LA/phenol molar ratio, LA/catalyst weight ratio, and time were optimized (60 °C, 1/4.5, 0.3, and 30 h, respectively), thus improving the *p*,*p*′-DPA yield up to 93 mol% without any marked drop in selectivity (entry 77, Table 6). At the end of the reaction, the ionic liquid was regenerated through ethyl acetate extraction, and it was recycled up to four runs. Only a slight decrease in the *p*,*p*′-DPA yield up to the value of 85 mol% was observed, which was mainly attributed to some loss of the ionic liquid that occurred during the separation step. Mthembu et al. [33] employed [EMIM][OTs] and [BMim]HSO_4_, considering the catalysis with H_2_SO_4_ (entry 5, Table 2), *p*-TSA (entry 9, Table 2), and CH_3_SO_3_H (entry 11, Table 2), for comparison purposes. Maximum DPA yields of 59 and 68 mol% were obtained with such ionic liquids (entries 78 and 79, Table 6), which are comparable to those already reported for the homogeneous catalysts. In addition, Liu et al. [22] tested several ionic liquids with Brønsted acidity (for which the corresponding chemical structures are reported in Appendix A), highlighting an unclear relationship between their acid strength and the corresponding catalytic activity, which decreased in the order of 4b > 1a > 4a = 2 > 5 > 6 > 3 > 1c > 1d (entries 80–88, Table 6). Remarkably, the highest *p*,*p*′-DPA yield of 74 mol% was achieved with the ionic liquid 4b, which included a thiol group. Exploiting again the beneficial effect of the thiol group on this condensation reaction, the authors proposed the simultaneous use of mercaptoacetic acid and the most interesting ionic liquids (namely, 1a, 2, 3, 4a, 5, and 6) (entries 89–94, Table 6). The claimed results confirmed the positive role of thiol for the improvement of the *p*,*p*′-DPA yield, which increased up to 71–84 mol% depending on the employed ionic liquid. However, at the same time, a decrease in the *p*,*p*′-DPA/*o*,*p*′-DPA molar ratio was ascertained, which seems in disagreement with the findings reported in the literature. In any case, the thiol effect must be considered not only in terms of the thermodynamic aspects but also kinetic ones, and it should be better exploitable at lower LA conversions [31]. Due to the presence of the thiol group in the 1b and 4b ionic liquids, these were tested as additives together with the 4a ionic liquid as the main acid catalyst (entries 95 and 96, Table 6). The authors claimed further improvement of the *p*,*p*′-DPA yield and *p*,*p*′-DPA/*o*,*p*′-DPA molar ratio, proving that thiol-containing anions of ionic liquid play a key role in the optimization of *p*,*p*′-DPA synthesis.

According to the above data, the fine tunability of the catalytic properties of the ionic liquids certainly offers remarkable advantages for improving DPA synthesis by exploiting the Brønsted sulfonic groups to promote the LA conversion according to the general mechanism already discussed for the other Brønsted acids (Figure 2). Moreover, the inclusion of a thiol group within the anion of the ionic liquid remarkably improved both the yield and selectivity of the *p*,*p*′-DPA. Although mechanistic details have not been provided by the authors, the improved selectivity of the *p*,*p*′-DPA isomer could be attributed to the improved cooperation between the thiol group and its cationic counterpart in the ionic liquid. However, the main bottleneck limiting the use of ionic liquids for such industrial applications is their high cost in addition to their uneasy isolation/reuse, which requires an additional separation step of the reaction mixture using an appropriate solvent extraction given the high boiling points of the involved compounds.

## 3. Applications of DPA: Challenges and Opportunities

The presence of one carboxyl group and two phenolic ones makes DPA an interesting molecule to be exploited for the synthesis of a plethora of more added-value bio-products. On the other hand, the reactivity of DPA must be properly controlled to avoid unwanted reactions, including its self-condensation [29,38]. Moreover, it can undergo oxidation and degradation over time [39,40], which can limit its shelf life and usability in several applications, thus requiring careful attention to its stability during the synthesis of the DPA-derived polymers; for instance, when transforming the carboxyl group of DPA in esters, ethers, or amide derivatives [41,42]. To date, DPA has been exploited for the synthesis of epoxy and phenolic resins [6,43] as well as other polymers, such as polycarbonates [44,45], polyarylates [46], and polyesters [47]. Both DPA-based epoxy and phenolic resins find applications in the production of composites, adhesives, and coatings [48,49]. Some niche applications of DPA have been proposed, including the production of thermal paper for printers [50], coatings, and adhesives [51,52,53]. Moreover, DPA and its derivatives show good antioxidant, antiviral, and antibacterial properties, making them suitable for the development of valuable applications in the cosmetic, food, textile, and polymer sectors [27,33,54,55]. Such materials could be obtained from the diphenolate derivatives. In this context, alkyl levulinates (ALs) were proposed by Olson in 2001 [56] as a replacement for LA in the synthesis of new diphenolate esters (DPEs). Remarkably, the reaction between ethyl levulinate (EL) and phenol under conditions analogous to those used for LA enables the production of DPE, but only in low yields, thus highlighting a much more moderate reactivity of the ALs. However, when the reaction conditions were properly modified, very high yields of the DPE (95 mol%) were ascertained, which were even greater than those obtained when starting from LA (67 mol%). The most efficient method for the synthesis of DPE from EL was achieved by using concentrated sulfuric acid as the catalyst, followed by dilution with ethanol at the end of the reaction. The residual reactants were steam-distilled off the product, but no further details regarding synthesis or purification were reported. Pancrazzi et al. [57] performed the one-pot synthesis of DPEs, which involved the LA’s esterification to EL and the cascade condensation of the latter with phenol and *ortho*-substituted phenols (R = 2-Me, 2-iso-Pr, 2-sec-Bu) without any intermediate purification step of ethyl levulinate. In principle, methyl and ethyl levulinate are the preferred ALs to be employed to exploit their highest esterification yields [58]. Therefore, two different approaches can be proposed for the synthesis of DPEs by employing ALs as starting materials (Figure 4).

The one-pot approach should be preferred over the two-step one because the former allows for a significant reduction in the work-up costs. Certainly, ALs should be preferred over LA for reducing the formation of by-products, especially when working under heterogeneous catalysis. Remarkably, the use of esters can open the way to the production of a new class of DPE-based polymers, such as bis epoxy and polyurethane derivatives [56], thus expanding the possibilities for applications with respect to the traditional DPA-based polymers. In the following paragraphs, these main applications will be discussed, with an emphasis on their relevant advantages and limitations over the corresponding BPA-based materials.

### 3.1. Epoxy Resins

DPA-based epoxy resins are employed for the synthesis of thermosetting materials, including coatings, adhesives, matrix resins, and others [48,59]. The presence of DPA within epoxy resins confers remarkable valuable properties, including lightweight and high-strength composites [60]. Moreover, epoxy resins containing DPA could be used for encapsulating and protecting electronic devices (semiconductors, printed circuit boards, and sensors), thus valorizing their excellent electrical insulation properties, thermal stability, and protection against moisture and contaminants [48,61,62]. Furthermore, epoxy resins are used to create molds and patterns for casting various materials, including plastics and metals, with the latter showing promising dimensional stability and heat resistance [60].

In deepening the exploitation possibilities of DPA, it is well-known that the synthesis of BPA glycidyl ethers (DGEBA) represents valuable polymers due to their robust mechanical properties, favorable thermodynamic characteristics, and remarkable stability [63]. DGEBA and its oligomers account for 90% of the global production of epoxy resins [64]. A similar approach to that of BPA can also be proposed for DPA, thus obtaining the corresponding glycidyl ethers (DGEDPA). Both chemical structures of the corresponding glycidyl ether monomers are reported in Figure 5.

The research towards the development of new DGEDPA-based materials is mainly aimed at solving some well-known DGEBA performance drawbacks, including high brittleness and poor impact resistance, due to its inherent rigid aromatic rings [65]. DGEDPA-based epoxy thermosets characterized by a hyperbranched structure may enhance mechanical strength and toughness simultaneously [66]. The hyperbranched structure generated through the reactivity of the carboxyl group of the DPA can significantly increase the crosslink density in thermosets, thereby enhancing their mechanical strength. This crosslinking could be obtained through the synthesis of ethers, esters, and amide derivatives of the DPA, thus exploiting the reactivity of its carboxylic group. In addition, hyperbranched polymers have a higher proportion of free volume than their linear counterparts, which helps to improve the toughness of DPA-based epoxy resins [67]. In this context, Wei et al. [42] synthesized a new DGEDPA-based polymer starting from DPA and epichlorohydrin to obtain a trifunctional epoxy resin. The obtained DGEDPA was cured with 3,3′-dimethyl-4,4′-diaminodicyclohexylmethane, and the final material exhibited better performance than that of the DGEBA-based one, including higher curing temperature, glass transition temperature, and thermal expansion characteristics [42]. Consequently, the authors claimed that this innovative epoxy resin shows promising potential for various critical applications, including aerospace and microelectronics. Regarding the electronic applications, McMaster et al. [68] tested a series of diglycidyl ethers of diphenolate esters (DGEDPE) as dielectric materials to be proposed in substitution of DGEBA. Methyl, ethyl, propyl, and butyl esters of DPA were synthesized and studied. Remarkably, the DGEDP-propyl showed the highest dielectric constant, with results comparable to those of the traditional DGEBA. The authors stated that variations in the dielectric characteristics of DGEDP-esters result from the complex interplay between segmental, local, and side-chain movements on one side and the influence of free volume and steric hindrance on the other. The research formed the groundwork for the development of new DGEDPE-based epoxy resins, with the intention to improve polarization and dielectric constants, which are valuable physical properties for such types of applications. Developing a different approach, Xiao et al. [65] synthesized different DPA-based hyperbranched epoxides (HBEPs) from DPA and dibromobutane to improve the mechanical properties of the final resin. A two-step synthetic approach was proposed by the authors (Figure 6). In the first step, hyperbranched polymers (HBPs) were synthesized through polymerization through a condensation mechanism starting from DPA and 1,4-dibromo butane in the presence of potassium carbonate and DMF. In the second step, the epoxidation reaction of HBPs to obtain HBEPs was carried out in the presence of an excess of epichlorohydrin. In this way, terminal functional groups were modified through epoxidation through epichlorohydrin, thus obtaining the epoxy resin. This methodology utilizes DPA as the branching component, thus improving the mechanical durability due to its aromatic ring composition and adding a well-tuned flexibility by including dibromo butane. Such hyperbranched thermosets exhibited exceptional mechanical strength and toughness, with much better performances than those of commercial DGEBA thermosets [65,69].

Qian et al. [41] proposed another innovative synthesis of a new class of epoxy resin with outstanding thermal performance, DPA-based polybenzoxazines, based on the amidation of DPA. First, the authors synthesized DPA hexanediammonium salt (DHDA) by reacting DPA and 1,6-hexanediamine in ethanol. Then, DHDA reacts with PEG-200, furfurylamine, and paraformaldehyde to obtain benzoxazine diphenolic hexanediamide (DHDA-fa) (Figure 7).

Polybenzoxazines offer distinct advantages compared to most conventional polymers. These advantages include minimal volumetric changes during curing, high glass transition temperatures, impressive thermal and mechanical characteristics, exceptional flame resistance, and a low dielectric constant [70]. The authors demonstrated that the incorporation of amide groups into a benzoxazine molecule can effectively improve the thermomechanical performances of these resins with respect to those of DPA-derived benzoxazines containing the free-ending carboxylic group in the DPA structure [41].

### 3.2. Phenolic Resins

Phenolic resins are well-known for their heat resistance and durability, making them suitable for producing electrical and automotive components, as well as in the construction industry [71]. DPA has been employed for the synthesis of phenolic resins, including, in particular, novolaks [72,73]. In this context, Kiran et al. [74] synthesized a sulfonated poly(arylene ether sulfone) copolymer through direct copolymerization of *p*,*p*′-DPA, benzene 1,4-diol, and the synthesized sulfonated 4,4′-difluorodiphenylsulfone (Figure 8).

The copolymer was subsequently crosslinked with 4,4′-(hexafluoroisopropylidene)diphenol epoxy resin through a thermal curing reaction to synthesize crosslinked membranes for fuel cell applications. Zúñiga et al. [75] claimed the synthesis of a DPA-based porous polybenzoxazine as the phenolic resin. For this purpose, the foaming agent (CO_2_) was generated in situ during the thermal curing. The DPA-benzoxazine monomer was thermally polymerized at different temperatures according to the reaction in Figure 9. According to the authors, the proposed synthesis is smart, as it uses the same chemical both as a crosslinking monomer and as a blowing agent. Moreover, the relative decarboxylation reaction can be controlled by varying the temperature, which gives an excellent and simple method for tuning foam properties. The obtained low-density foams exhibit a high glass transition temperature, with heterogeneous open-cell morphology.

An alternative to DPA-based sulfonated poly(arylene ether sulfone) copolymers and porous polybenzoxazine is represented by DPE-based resins. To improve the thermal stability of aromatic polyesters and polycarbonates prepared from DPA, various DPEs have been proposed for the synthesis of new polymers, including methyl diphenolate (MDP), ethyl diphenolate (EDP), *n*-propyl diphenolate (PDP), *n*-butyl diphenolate (BDP), and *n*-hexyl diphenolate (HDP) [44,76]. For example, Ping et al. [76] synthesized two aromatic polyesters, poly(MDP–IPC) and poly(EDP–IPC), via interfacial polycondensation (IPC) of isophthaloyl chloride with methyl diphenolate (MDP) and ethyl diphenolate (EDP), respectively, and studied their thermal stability by considering the poly(DPA–IPC) as the reference. The authors found that the presence of carboxylic groups in poly(DPA–IPC) leads to lower thermal stability addressed to carboxyl group interactions, which can be weakened by working with the ester. In this context, poly(MDP–IPC) and poly(EDP–IPC) display single-stage thermal decomposition at higher temperatures, with slightly reduced glass transition temperatures due to increased side-chain flexibility. Overall, at the state of the art, DPE-based polymers exhibit better thermal performance and behavior than DPA-based ones.

### 3.3. Thermal Papers

DPA could also be used as a sustainable alternative to BPA to produce thermal papers, which are commonly used in point-of-sale (POS) receipt, fax, and label printers [50]. Thermal papers enable the production of images after exposure to a heat source, thus eliminating the need for the use of inks or toners. Generally, the thermosensitive layer contains a fluoran-type dye (leuco dye) serving as the thermal dye and a color developer, such as BPA, as the proton donor. These chemical components are responsible for color development after heating, according to Figure 10.

The U.S. Environmental Protection Agency (EPA) has issued warnings about using BPA on thermal paper and is searching for possible substitutes, including DPA-based ones [77]. Moreover, DPA and its derivatives were proposed as drying agents for inkjet receptor media, as reported by Malhotra et al. [78]. In this context, when DPA is used in combination with a multivalent salt and a surfactant, it can efficiently dehydrate the medium, resulting in a better quality of the final printed product.

### 3.4. Coatings and Adhesives

DPA also finds applications for the formulation of coatings and adhesives [51,52,53]. To date, most of these applications of DPA are aimed at improving adhesion promotion, corrosion resistance, thermal stability, resin modification, adhesive strength, electrical insulation, and UV resistance [79]. Remarkably, Yan et al. [52] first synthesized the poly(diphenolic acid-phenyl phosphate) [poly(DPA-PDCP)] from DPA (Figure 11) and then used it in combination with a polyethylenimine (PEI) to produce, through layer-by-layer self-assembly, a flame-retardant surface coating for ramie fabric. Poly(DPA-PDCP)/PEI showed excellent flame-retardant properties, thus finding many potential applications in the automobile, train, and aerospace fields [53].

Furthermore, DPA is effective for improving the UV resistance of coatings by helping to prevent color fading and degradation when exposed to sunlight [79,80]. This is a valuable property for outdoor applications, such as architectural coatings and automotive finishes. For example, in the study by Zhang et al. [79], a novel, multifunctional, bioderived polyphosphate (PPD) for poly(lactic acid) (PLA) was synthesized through a three-step procedure (Figure 12). In the first step, 9,10-dihydro-9-oxa-10-phosphaphanthrene-10-oxide (DOPO) reacted with polyformaldehyde (HCHO) to produce a DOPO-derived compound, 2-(6-oxido-6H-dibenz[*c*,*e*][1,2]oxaphosphorin-6-yl)-methanol (ODOPM). ODOPM reacted with DPA in the presence of N, N-dimethylformamide (DMF) and sulfuric acid (as the acid catalyst) to produce (6-oxidodibenzo[*c*,*e*]› [1,2]oxaphosphinin-6-yl)methyl 4,4-bis(4-hydroxyphenyl)pentanoate (DPOM). Lastly, DPOM reacted with phenylphosphonic dichloride (PPDC) in ethyl acetate (EAC) and triethylamine (Et_3_N) to give the target PPD.

### 3.5. Antioxidant Properties

DPA shows interesting antioxidant properties, which could potentially be exploited for various applications, such as in the packaging industry to extend the shelf life of products and in cosmetics for their anti-ageing effects. Moreover, DPA and its derivatives can be used in paint formulations, decorative surface coatings, additives for lubricating oil, plasticizers, surfactants, cosmetics, and the textile industry [27,54,55,81,82]. For example, in the rubber industry, DPA is employed as an antioxidant additive, thus helping to protect such products from the degradation of oxygen, heat, and other agents [83]. This can extend the lifespan of rubber components used in automotive and industrial applications. Similarly, in the paint industry, the addition of DPA derivatives can increase the durability of commercial paint formulations. As reported in the patent of Holmen et al. [82], such an improvement was achieved through the incorporation of a DPA-based polyamide. Even at the low amount of 1 wt% of DPA within the polyamide modifier, this formulation improves the longevity of an applied coating. In all of these applications, DPA serves as a valuable antioxidant additive by effectively scavenging free radicals and inhibiting the oxidation of materials, thereby extending the lifespan, maintaining performance, and ensuring the quality of various products and materials.

### 3.6. Other Possible Applications

Finally, DPA-coated surfaces show good antivirus and antibacterial properties. This valuable biological activity is attributed to the interaction between the DPA molecules and the membrane proteins of the pathogens, thus causing deformation of the cell membrane and the cell lysis [84]. In this context, Shen et al. [84] demonstrated the active biological activity of Co-DPA-1.0% fabric surface towards common pathogens, like *Escherichia coli* and *Staphylococcus aureus*. DPA can be employed as an antibacterial agent in textile finishes and coatings to enhance the resistance of textiles to biological factors, which can cause fading and degradation. For example, Shen et al. [84] employed DPA as a finishing reagent to impart antibacterial and antiviral functions to cotton fabric. After a simple pad–dry–cure process, DPA molecules were covalently linked onto cotton fiber surfaces via the esterification reaction between their carboxyl groups and the hydroxyl groups of cellulose on the fiber surfaces. The DPA phenolic groups on the cotton fibers enable the destruction of the pathogens through protein affinity interactions. Experimental results show that the DPA-modified fabrics realize not only a high bacteriostatic effect against both *Escherichia coli* and *Staphylococcus aureus* but also an excellent antivirus effect that allows for rapid virus inactivation in less than 20 min. It was also demonstrated that the modification process generated insignificant damage to the cotton fiber structure, and the resultant cotton fabrics are safe for human skin. Therefore, this work may open a new way to endow cotton textiles with antibacterial and antiviral functions. Moreover, some phenolic acids have been studied for their potential biological activity, including anti-inflammatory and anti-cancer properties [85,86], thus opening the way to the development of new therapeutic applications.

## 4. Conclusions

This review focused on the synthesis and potential applications of diphenolic acid, a valuable alternative to bisphenol A, and demonstrated that its synthesis from the condensation of levulinic acid and phenol still has many challenges and critical issues to be solved. The feasibility of this reaction has been variously investigated, requiring, similarly to bisphenol A, the use of a Brønsted acid catalyst, mild temperature, and long reaction time. Mineral acids give good results in terms of yield (about 70 mol%) at complete levulinic acid conversion, but the development of heterogeneous acid catalysts is highly desirable, thus minimizing the corrosion problems and simplifying the subsequent work-up operations. Commercial acid resins, such as Amberlyst-15 and zeolites, give promising catalytic performances similar to those of traditional homogeneous acids. Moreover, as discussed in detail, new ad hoc catalytic systems have been proposed by several authors, which, in some cases, maintain good catalytic performances after the reactivation step. In fact, catalyst deactivation was generally observed after its recycling mainly due to the adsorption of organic matter on the catalyst surface. On this basis, the thermal calcination of the spent catalyst is often effective to restore its pristine performances. However, this thermal treatment is feasible only for inorganic systems (sulfonated N-doped carbon nanotube, H_3_PW_12_O_40_-silica materials, and zeolites), whilst it is inappropriate for sulfonic-acid-based organic resins (Amberlyst and Nafion), which would degrade at such high temperatures. Therefore, the inorganic catalysts are industrially more attractive from the perspective of a desirable scale-up of the reaction in the immediate future. The improvement of the regioselectivity of this reaction is specifically addressed because different isomers, including, mainly, the *p*,*p*′- and *o*,*p*′-ones, can be obtained starting from phenol, but only the first one is industrially attractive. The most studied solution to solve this issue involves the use of thiol additives, which mainly improve the *p*,*p*′/*o*,*p*′ molar ratio, in some cases up to 100, which allows for reaching a *p*,*p*′-DPA yield higher than 90 mol%. With these kinds of catalytic systems, the thiol effect should be properly tuned by preferring a kinetic control regime and therefore working at limited levulinic acid conversion rather than a thermodynamic one. In most cases, this last approach has been preferred by pursuing the maximum performance of the catalysts and thus mostly losing the advantage deriving from the thiol use. Remarkably, only batch-scale studies have been carried out up to now, whilst thiol use could be even better exploited through a continuous-scale approach. In addition to the catalytic synthesis of DPA, its main developed applications have also been discussed, which mainly deal with epoxy and phenolic resins and polycarbonates. Moreover, DPA shows many other potential valuable benefits in terms of antioxidant, flame-retardant, and UV protection properties and biological activity. In principle, the work-up step should be specifically evaluated depending on the use of the final material. In this context, some old patents for DPA synthesis reported some purification procedures, but the evaluation of DPA purity is limited by the technological techniques of those years in the field of characterization. Nowadays, the phase diagrams of the *p*,*p*′- and *o*,*p*′-isomers are not known, and the characterization of the heavier by-products has been little investigated. Therefore, further research on these issues is highly necessary to fill these gaps and improve the know-how mainly in the selective synthesis of the *p*,*p*′-isomer and the industrial separation/purification field. Lastly, other challenges to be further investigated include the use of substituted phenolic derivatives, which can be reasonably obtained from lignin sources, and/or the use of alkyl levulinates instead of levulinic acid to increase the selectivity to the *p*,*p*′-isomer and improve the physico-chemical properties of the final polymers. Very few promising results deriving from these approaches are already available, demonstrating that further improvement in this field is possible and thus favoring the growth of the production of DPA and its added-value, versatile derivatives.

## Figures and Tables

**Figure 1 molecules-29-00126-f001:**
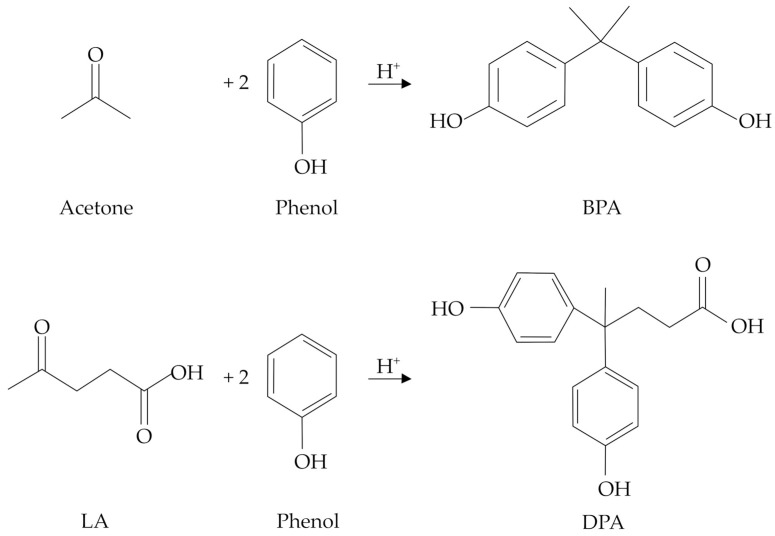
Acid-catalyzed syntheses of BPA and DPA.

**Figure 2 molecules-29-00126-f002:**
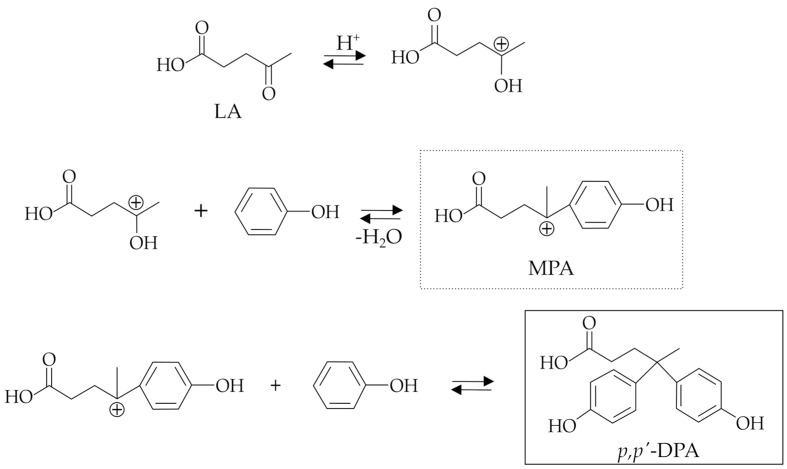
General reaction mechanism proposed for the synthesis of DPA (adapted from [16]).

**Figure 3 molecules-29-00126-f003:**
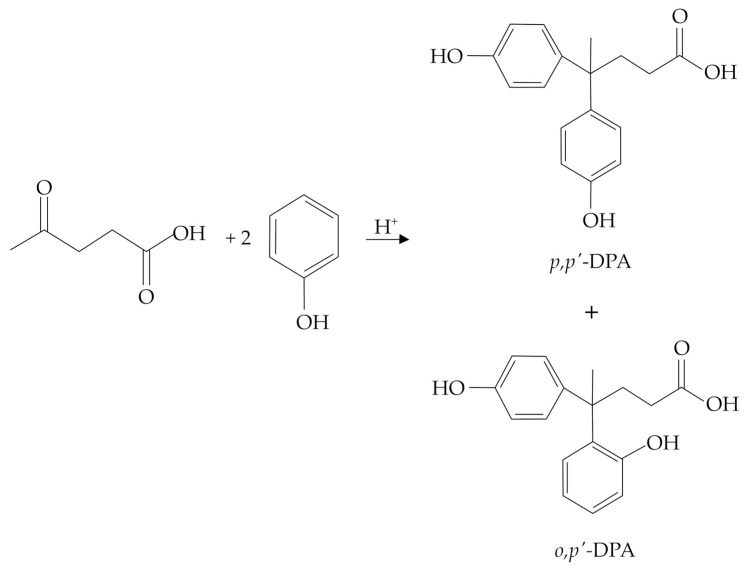
Main DPA isomers originating from the acid-catalyzed reaction occurring between LA and phenol.

**Figure 4 molecules-29-00126-f004:**
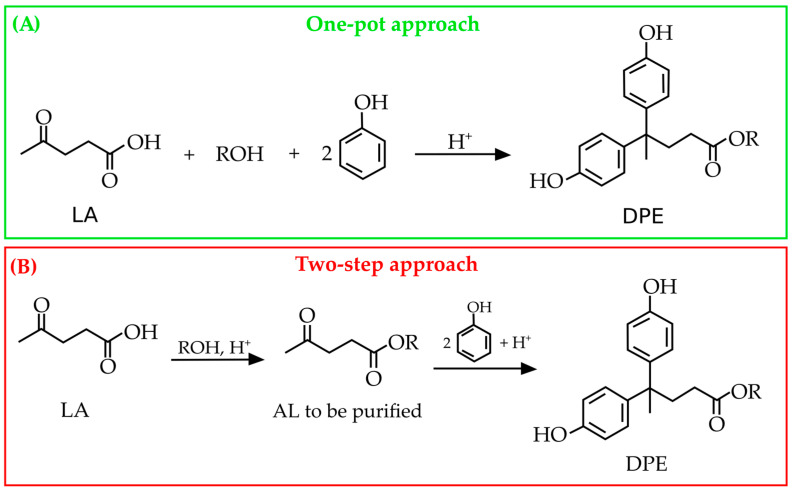
Comparison between the one-pot approach (**A**) and the two-step one (**B**) for the synthesis of DPE derivatives from ALs according to Pancrazzi et al. [57].

**Figure 5 molecules-29-00126-f005:**
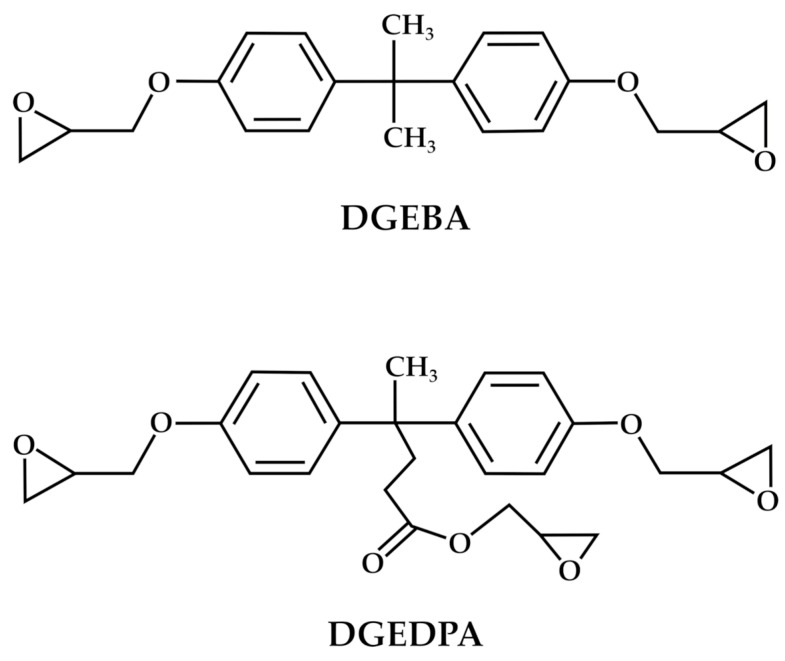
Comparison between the chemical structures of DGEBA and DGEDPA.

**Figure 6 molecules-29-00126-f006:**
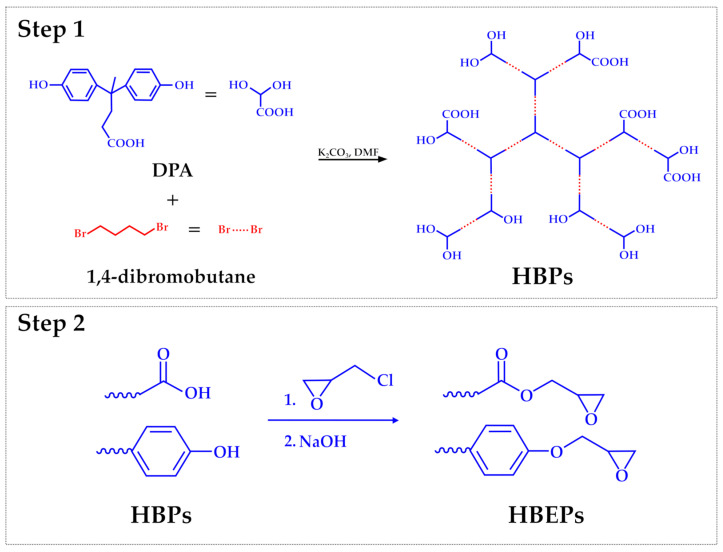
Synthesis of HBEPs, according to Xiao et al. [65].

**Figure 7 molecules-29-00126-f007:**
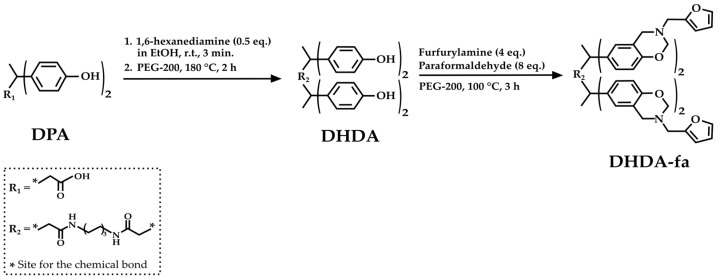
Chemical structure of DPA-derived benzoxazine diphenolic hexanediamide, according to Qian et al. [41].

**Figure 8 molecules-29-00126-f008:**
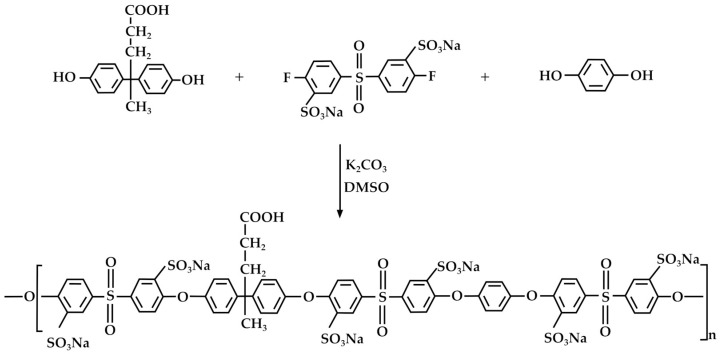
Synthesis of sulfonated poly(arylene ether sulfone) copolymer starting from DPA, according to Kiran et al. [74].

**Figure 9 molecules-29-00126-f009:**
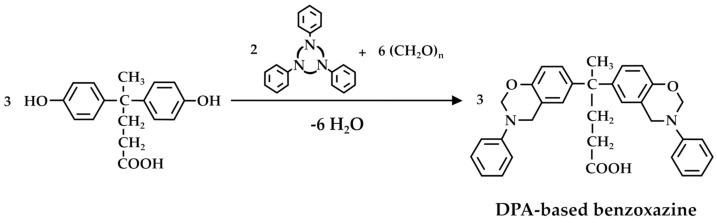
Synthesis of DPA-based polybenzoxazines according to Zúñiga et al. [75].

**Figure 10 molecules-29-00126-f010:**
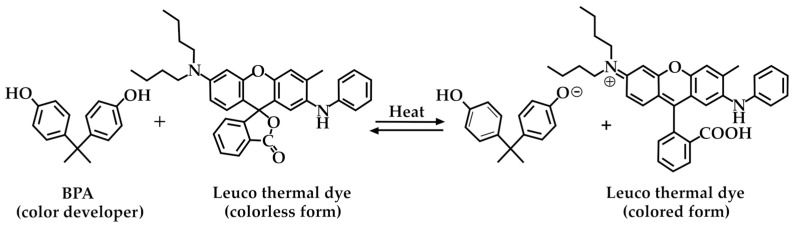
Reaction between BPA (color developer), and the leuco thermal dye (colorless) to give the colored open form of the dye [50].

**Figure 11 molecules-29-00126-f011:**
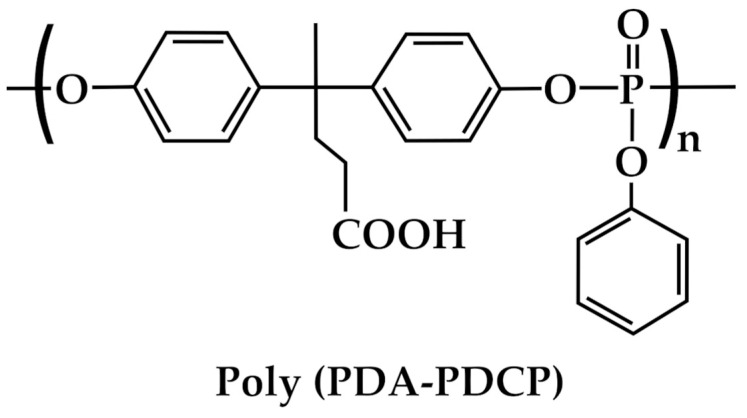
Structure of the [poly(DPA-PDCP)].

**Figure 12 molecules-29-00126-f012:**
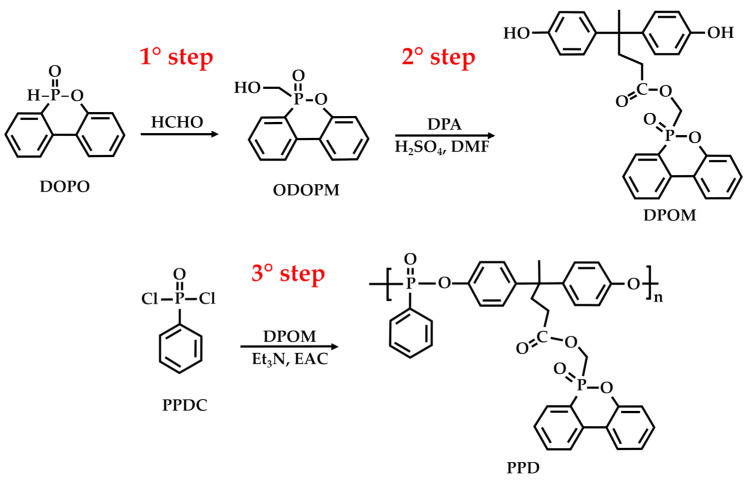
Synthesis of PPD, according to Zhang et al. [79].

**Table 1 molecules-29-00126-t001:** Physico-chemical properties of DPA.

Physical Property	Details
Formula	C_17_H_18_O_4_
Chemical structure	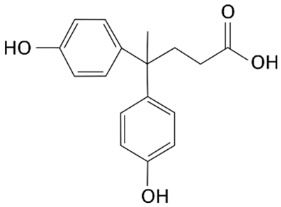
Molecular weight (g mol^−1^)	286.33
Melting point (°C)	167–170
Boiling point (°C)	507
Flash point (°C)	208
Density (g mL^−1^)	1.30–1.32
Refractive index	1.675
Solubility	Slightly soluble in water; soluble in acetic acid, acetone, ethanol, isopropanol, and methyl ethyl ketone; and insoluble in benzene

**Table 2 molecules-29-00126-t002:** DPA synthesis catalyzed using inorganic mineral acids and homogeneous sulfonic acids.

Entry	LA/Phenol (mol/mol)	Catalyst (g_LA_/g_cat_)	Additive (mol_LA_/mol_additive_)	T (°C)	t (h)	C_LA_ (mol%)	Y*_p,p_*_′-DPA_ (mol%)	*p*,*p*′-DPA/*o*,*p*′-DPA Molar Ratio	Ref.
1	1/3	HCl (8.0)	/	100	6	46	19	2.2	[18]
2	1/7	HCl (8.0)	/	100	24	n.a. ^a^	56	1.8	[20]
3	1/4	HCl (6.3)	Ethanethiol (100)	60	24	65	41	2.0	[21]
4	1/4	H_2_SO_4_ (11.8)	/	60	24	62	42	11.2	[19]
5	1/3.7	H_2_SO_4_ (0.9)	/	75	6	n.a. ^a^	74 ^b^	n.a. ^a^	[33]
6	1/4	H_2_SO_4_ (2.3)	/	60	48	n.a. ^a^	61	24.0	[22]
7	1/4	H_2_SO_4_ (2.3)	Mercaptoacetic acid (20.0)	60	48	n.a. ^a^	68	9.0	[22]
8	1/4	*p*-TSA (1.3)	Ethanethiol (100)	60	24	25	18	3.9	[21]
9	1/3.7	*p*-TSA (0.9)	/	75	6	n.a. ^a^	70 ^b^	n.a. ^a^	[33]
10	1/4	NH_2_SO_3_H (2.4)	Ethanethiol (100)	60	24	58	51	7.4	[21]
11	1/3.7	CH_3_SO_3_H (0.9)	/	75	6	90	37(66 ^b^)	1.2	[33]
12	1/9.2	CH_3_SO_3_H (0.9)	/	75	6	n.a. ^a^	86 ^b^	n.a. ^a^	[33]
13	1/4	CF_3_SO_3_H (1.5)	/	60	48	n.a. ^a^	43	25.0	[22]
14	1/4	HS(CH_2_)_3_SO_3_H (1.5)	/	60	48	n.a. ^a^	77	30.0	[22]

^a^ n.a. = not available; ^b^ yield of *p*,*p*′-DPA + *o*,*p*′-DPA.

**Table 3 molecules-29-00126-t003:** Heterogeneous catalysts with sulfonic acids applied to the synthesis of DPA.

Entry	LA/Phenol (mol/mol)	Catalyst (g_LA_/g_cat_)	Additive (mol_LA_/mol_additive_)	T (°C)	t (h)	C_LA_ (mol%)	Y*_p,p_*_′-DPA_ (mol%)	*p*,*p*′-DPA/*o*,*p*′-DPA Molar Ratio	Ref.
15	1/4	Amberlyst-15 (6.1)	/	120	24	64	6	0.4	[27]
16	1/4	Amberlyst-15 (4.4)	/	60	24	70	55	15.8	[19]
17	1/3	Amberlyst-15 (9.0)	/	100	16	34	14	4.0	[30]
18	1/3	Nafion NR50 (1.7)	/	100	16	36	24	5.8	[30]
19	1/3	SHPAOs ^a^ (7.6)	/	100	16	40	35	7.6	[30]
20	1/3	SHPAOs ^a^ (7.6)	Benzylthiol (15.5)	100	16	65	42	15.6	[30]
21	1/3	SHPAOs ^a^ (7.6)	Ethanethiol (15.5)	100	16	70	53	19.5	[30]
22	1/3	SHPAOs ^a^ (7.6)	1-Propanethiol (15.5)	100	16	65	49	17.6	[30]
23	1/3	SHPAOs ^a^ (7.6)	1-Butanethiol (15.5)	100	16	60	38	14.0	[30]
24	1/3	SHPAOs ^a^ (7.6)	2-Propanethiol (15.5)	100	16	54	38	12.0	[30]
25	1/3	SHPAOs ^a^ (7.6)	2-Methyl-2-propanethiol (15.5)	100	16	39	26	10.5	[30]
26	1/3	SHPAOs ^a^ (7.6)	1-Butanethiol (3.9)	100	32	93	n.a. ^d^	20.0	[31]
27	1/3	SHPAO_s_-MEA ^b^ (8.2)	/	100	16	59	38	9.8	[30]
28	1/3	SHPAOs-TEP ^c^ (10.1)	/	100	16	57	35	15.5	[30]
29	1/4	Fe@NC-SO_3_H ^d^ (1.2)	/	60	24	69	49	17.5	[19]
30	1/4	Ni@NC-SO_3_H ^d^ (1.9)	/	60	24	72	57	17.2	[19]
31	1/4	Co@NC-SO_3_H ^d^ (2.1)	/	60	24	74	63	16.9	[19]
32	1/4	Co@NC-SO_3_H ^d^ (2.1)	Mercaptoacetic acid (5.0)	60	24	82	76	24.4	[19]
33	1/4	Co@NC-SO_3_H ^d^ (2.1)	Mercaptoacetic acid (5.0)	60	48	98	91	23.7	[19]

^a^ Sulfonated hyperbranched poly(arylene oxindole)s; ^b^ Sulfonated hyperbranched poly(arylene oxindole)s modified with 2-mercaptoethylamine; ^c^ Sulfonated hyperbranched poly(arylene oxindole)s modified with 4-(2-thioethyl)-pyridine; ^d^ Sulfonated N-doped carbon nanotube.

**Table 4 molecules-29-00126-t004:** Heteropolyacids applied to the synthesis of DPA.

Entry	LA/Phenol (mol/mol)	Catalyst (g_LA_/g_cat_)	T (°C)	t (h)	C_LA_ (mol%)	Y*_p_*_,*p*′-DPA_ (mol%)	*p*,*p*′-DPA/*o*,*p*′-DPA Molar Ratio	Ref.
34	1/4	H_3_PW_12_O_40_ (4.0)	100	6	55	46 ^a^	n.a. ^b^	[16]
35	1/4	H_4_SiW_12_O_40_ (4.0)	100	6	69	25 ^a^	n.a. ^b^	[16]
36	1/4	H_4_SiMo_12_O_40_ (6.4)	100	6	79	17 ^a^	n.a. ^b^	[16]
37	1/4	H_3_PW_12_O_40_ (4.0)	140	6	87	82(85 ^a^)	28.0	[16]
38	1/7	H_3_PW_12_O_40_ (7.9)	100	24	n.a. ^b^	1	2.4	[20]
39	1/3	H_3_PW_12_O_40_ (4.2)	100	16	55	31	8.7	[30]
40	1/3	H_3_PW_12_O_40_ (8.0)	100	6	33	18	1.5	[18]
41	1/3	H_6_P_2_W_18_O_62_ (8.0)	100	6	39	29	3.5	[18]
42	1/4	H_3_PW_12_O_40_/SiO_2_-E-4.0 ^c^ (8.0)	100	8	24	1	3.6	[32]
43	1/4	H_3_PW_12_O_40_/SiO_2_-E-7.5 ^c^ (8.0)	100	8	31	2	2.8	[32]
44	1/4	H_3_PW_12_O_40_/SiO_2_-E-14.5 ^c^ (8.0)	100	8	75	21	3.0	[32]
45	1/4	H_3_PW_12_O_40_/SiO_2_-E-17.5 ^c^ (8.0)	100	8	80	27	2.8	[32]
46	1/4	H_3_PW_12_O_40_/SiO_2_-C-7.5 ^c^ (8.0)	100	8	19	1	2.1	[32]
47	1/7	H_3_PW_12_O_40_/SiO_2_-3D_hex_-C-7.5 ^d^ (7.9)	100	24	n.a. ^b^	4	1.6	[20]
48	1/7	H_3_PW_12_O_40_/SiO_2_-3D_hex_-C-11.1 ^d^ (7.9)	100	24	n.a. ^b^	14	1.8	[20]
49	1/7	H_3_PW_12_O_40_/SiO_2_-3D_hex_-C-14.5 ^d^ (7.9)	100	24	n.a. ^b^	13	1.3	[20]
50	1/7	H_3_PW_12_O_40_/SiO_2_-3D_hex_-C-15.8 ^d^ (7.9)	100	24	n.a. ^b^	12	1.1	[20]
51	1/7	H_3_PW_12_O_40_/SiO_2_-3D_hex_-C-29.9 ^d^ (7.9)	100	24	n.a. ^b^	12	1.7	[20]
52	1/7	H_3_PW_12_O_40_/SiO_2_-3D_hex_-C-35.2 ^d^ (7.9)	100	24	n.a. ^b^	9	1.7	[20]
53	1/7	H_3_PW_12_O_40_/SiO_2_-3D_hex_-C-43.6 ^d^ (7.9)	100	24	n.a. ^b^	9	1.7	[20]
54	1/7	H_3_PW_12_O_40_/SiO_2_-2D_hex_-C-11.7 ^d^ (7.9)	100	24	n.a. ^b^	3	3.3	[20]
55	1/7	H_3_PW_12_O_40_/SiO_2_-2D_hex_-E-11.1 ^d^ (7.9)	100	24	n.a. ^b^	6	3.3	[20]
56	1/3	Cs_1.5_H_4.5_P_2_W_18_O_62_ (8.0)	100	6	36	31	7.3	[18]
57	1/3	Cs_2.5_H_0.5_PW_12_O_40_ (8.0)	100	6	28	22	4.0	[18]
58	1/9	Cs_1.5_H_4.5_P_2_W_18_O_62_ (4.0) ^e^	150	10	n.a. ^b^	62	7.3	[18]
59	1/4	Cs_2.5_H_0.5_PW_12_O_40_ (4.0) ^e^	150	10	n.a. ^b^	37	4.9	[18]

^a^ Yield of *p*,*p*′-DPA + *o*,*p*′-DPA; ^b^ n.a.= Not available; ^c^ Calcined(C)/extracted(E)H_3_PW_12_O_40_-silica materials with x wt% of H_3_PW_12_O_40;_
^d^ Hexagonal(hex) calcined(C)/extracted(E) 3D/2D H_3_PW_12_O_40_-silica materials with x wt% of H_3_PW_12_O_40_; ^e^ The test was carried out with a stirring rate of 1200 rpm.

**Table 5 molecules-29-00126-t005:** Zeolites and modified metal oxides applied to the synthesis of DPA.

Entry	LA/Phenol (mol/mol)	Catalyst (g_LA_/g_cat_)	T (°C)	t (h)	C_LA_ (mol%)	Y*_p_*_,*p*′-DPA_ (mol%)	*p*,*p*′-DPA/*o*,*p*′-DPA Molar Ratio	Ref.
60	1/4	Pr-SO_3_H-SBA-15 ^c^ (6.1)	120	24	38	2	0.4	[27]
61	1/4	Ar-SO_3_H-SBA-15 ^d^ (6.1)	120	24	41	3	0.3	[27]
62	1/4	Nafion-SBA-15 (6.1)	120	24	38	5	0.5	[27]
63	1/4	n-ZSM-5 (6.1)	120	24	26	1	4.9	[27]
64	1/4	H-USY (6.1)	120	24	37	8	2.0	[27]
65	1/4	H-Beta 12.5 (6.1)	120	24	44	33	99	[27]
66	1/4	H-Beta 19 (6.1)	120	24	48	40 ^a^	n.a. ^b^	[27]
67	1/4	H-Beta 75 (6.1)	120	24	57	45 ^a^	n.a. ^b^	[27]
68	1/4	H-Beta 180 (6.1)	120	24	49	27 ^a^	n.a. ^b^	[27]
69	1/6	H-Beta 19 (4.6)	140	72	77	69	99	[27]

^a^ Yield of *p*,*p*′-DPA + *o*,*p*′-DPA; ^b^ n.a. = not available; ^c^ Propylsulfonic-acid functionalized mesostructured silica; ^d^ Arenesulfonic-acid functionalized mesostructured silica.

**Table 6 molecules-29-00126-t006:** Ionic liquids applied to the synthesis of DPA.

Entry	LA/Phenol (mol/mol)	Catalyst (g_LA_/g_cat_)	Additive (mol_LA_/mol_additive_)	T (°C)	t (h)	C_LA_ (mol%)	Y*_p_*_,*p*′-DPA_ (mol%)	*p*,*p*′-DPA/*o*,*p*′-DPA Molar Ratio	Ref.
70	1/4	[BSMim]CF_3_SO_3_ ^a^ (0.6)	Ethanethiol (100)	60	24	81	79	100	[21]
71	1/4	[BSMim]OAc ^b^ (0.8)	Ethanethiol (100)	60	24	35	34	100	[21]
72	1/4	[BSMim]HSO_4_ ^c^ (0.7)	Ethanethiol (100)	60	24	75	74	100	[21]
73	1/4	[BPy]HSO_4_ ^d^ (1.0)	Ethanethiol (100)	60	24	68	67	90	[21]
74	1/4	[AMim]Br ^e^ (1.1)	Ethanethiol (100)	60	24	18	10	3.1	[21]
75	1/4	[BMim]Cl ^f^ (1.3)	Ethanethiol (100)	60	24	14	11	4.6	[21]
76	1/4	[BSMim]CF_3_SO_3_ ^a^ (0.6)	/	60	24	73	70	22.3	[21]
77	1/4.5	[BSMim]HSO_4_ ^c^ (0.3)	Ethanethiol (100)	60	30	n.a. ^g^	93	100	[21]
78	1/3.7	[EMIM][OTs] ^h^ (0.9)	/	75	6	n.a. ^g^	59 ^i^	n.a. ^g^	[33]
79	1/3.7	[BMim]HSO_4_ ^j^ (0.9)	/	75	6	n.a. ^g^	68 ^i^	n.a. ^g^	[33]
80	1/4	1a (0.8) ^k^	/	60	48	70	52	16.0	[22]
81	1/4	1c (0.7) ^k^	/	60	48	n.a. ^g^	34	32.0	[22]
82	1/4	1d (0.6) ^k^	/	60	48	n.a. ^g^	13	30.0	[22]
83	1/4	2 (0.6) ^k^	/	60	48	n.a. ^g^	48	21.0	[22]
84	1/4	3 (0.7) ^k^	/	60	48	n.a. ^g^	43	33.0	[22]
85	1/4	4a (0.8) ^k^	/	60	48	n.a. ^g^	48	32.0	[22]
86	1/4	4b (0.8) ^k^	/	60	48	n.a. ^g^	74	50.0	[22]
87	1/4	5 (0.7) ^k^	/	60	48	n.a. ^g^	47	33	[22]
88	1/4	6 (0.7) ^k^	/	60	48	n.a. ^g^	46	33	[22]
89	1/4	1a (0.8) ^k^	Mercaptoacetic acid (20.0)	60	48	n.a. ^g^	71	9.0	[22]
90	1/4	2 (0.6) ^k^	Mercaptoacetic acid (20.0)	60	48	n.a. ^g^	72	9.0	[22]
91	1/4	3 (0.7) ^k^	Mercaptoacetic acid (20.0)	60	48	n.a. ^g^	78	9.0	[22]
92	1/4	4a (0.8) ^k^	Mercaptoacetic acid (20.0)	60	48	n.a. ^g^	84	20.0	[22]
93	1/4	5 (0.7) ^k^	Mercaptoacetic acid (20.0)	60	48	n.a. ^g^	73	13.0	[22]
94	1/4	6 (0.7) ^k^	Mercaptoacetic acid (20.0)	60	48	n.a. ^g^	73	9.0	[22]
95	1/4	4a (0.9) ^k^	1b (20.0)	60	48	n.a. ^g^	85	100	[22]
96	1/4	4a (0.9) ^k^	4b (20.0)	60	48	n.a. ^g^	91	100	[22]

^a^ 1-(4-sulphonic acid)butyl-3-methylimidazolium triflate; ^b^ 1-(4-sulphonic acid)butyl-3-methylimidazolium acetate; ^c^ 1-(4-sulphonic acid)butyl-3-methylimidazolium hydrogen sulphate; ^d^ 1-butylpyridinium hydrogen sulphate; ^e^ 1-allyl-3-methylimidazolium bromide; ^f^ 1-butyl-3-methylimidazolium chloride; ^g^ n.a. = not available; ^h^ 1-ethyl-3-methylimidazolium tosylate; ^i^ yield of *p*,*p*′-DPA + *o*,*p*′-DPA; ^j^ 1-butyl-3-methylimidazolium hydrogen sulphate; ^k^ chemical structure is reported in Appendix A.

## Data Availability

Not applicable.

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
