# Peer review of "Challenges and Opportunities in the Catalytic Synthesis of Diphenolic Acid and Evaluation of Its Application Potential"

_molecules, 2023, doi:10.3390/molecules29010126_

Round 1

Reviewer 1 Report

Comments and Suggestions for Authors

This work is a comprehensive critical review on the most updated progress on various processes that have been developed for the production of diphenolic acid (DPA) from levulinic acid. I really enjoyed reading it, and in my opinion it can provide insightful understanding of the recent progress, not only on the catalytic approaches but also on the potential industrial uses of DPA, providing helpful and useful information and guide for future research work. Noteworthy, it is not just a literature survey on the subject, but the authors have made an important effort in critically analyzing the importance, impact and reliability of published work. Additionally, in section 5 (Conclusions), they introduce interesting suggestions on the most critical issues, that will surely be interesting for the readers of Molecules. Therefore, I am glad to give my support for its publication.

I only have a couple of minor suggestions for the authors, aiming to complete or improve some aspects of the manuscript:

-          I miss section 4 in the text, somehow the authors pass from section 3.6 Other applications to section 5. Conclusions.

-          Section 2.4 is entitled DPA synthesis with other heterogeneous systems: zeolites and modified metal oxides. But the latter is not true, the analyzed catalysts are not usually considered modified metal oxides, since they are sulfonic acid-modified silicas. I recommend the authors to modify it to avoid confusion in the potential reader.

Author Response

This work is a comprehensive critical review on the most updated progress on various processes that have been developed for the production of diphenolic acid (DPA) from levulinic acid. I really enjoyed reading it, and in my opinion it can provide insightful understanding of the recent progress, not only on the catalytic approaches but also on the potential industrial uses of DPA, providing helpful and useful information and guide for future research work. Noteworthy, it is not just a literature survey on the subject, but the authors have made an important effort in critically analyzing the importance, impact and reliability of published work. Additionally, in section 5 (Conclusions), they introduce interesting suggestions on the most critical issues, that will surely be interesting for the readers of Molecules. Therefore, I am glad to give my support for its publication.

I only have a couple of minor suggestions for the authors, aiming to complete or improve some aspects of the manuscript:

Reviewer: I miss section 4 in the text, somehow the authors pass from section 3.6 Other applications to section 5. Conclusions.

Authors: We are sorry for this typo error and we have changed the number of Conclusions section to “4. Conclusions”.

Reviewer: Section 2.4 is entitled DPA synthesis with other heterogeneous systems: zeolites and modified metal oxides. But the latter is not true, the analyzed catalysts are not usually considered modified metal oxides, since they are sulfonic acid-modified silicas. I recommend the authors to modify it to avoid confusion in the potential reader.

Authors: We thank the Reviewer for this suggestion; thus we have modified the title of Section 2.4 to “DPA synthesis with other heterogeneous systems: zeolites and modified silicas”. We hope that now it could be much clear.

Reviewer 2 Report

Comments and Suggestions for Authors

The paper describes the review on “Challenges and opportunities in the catalytic synthesis of diphenolic acid and evaluation of its application perspectives”. The paper is very interesting well written and suitable for Molecules. It is suitable to publish as it is. Of course, if the authors of the manuscript can cite and summarize more literatures from the past three years, it will be more attractive to the readers of the manuscript. (Accepted)

Author Response

Reviewer: The paper describes the review on “Challenges and opportunities in the catalytic synthesis of diphenolic acid and evaluation of its application perspectives”. The paper is very interesting well written and suitable for Molecules. It is suitable to publish as it is. Of course, if the authors of the manuscript can cite and summarize more literatures from the past three years, it will be more attractive to the readers of the manuscript. (Accepted)

Authors: We thank the Reviewer for his/her appreciation of our manuscript. Regarding the other observation, the available literature is not so extensive and all the articles published from the past three years have been already included and discussed in the manuscript.

Reviewer 3 Report

Comments and Suggestions for Authors

This review deals with the synthesis and applications of diphenolic acid (DPA), considering that this molecule has emerged as a sustainable alternative for the replacement of bisphenol A with important toxicological problems. Review papers on a specific topic allow readers to have an updated vision of the progress made, and, in this sense, this paper tries to cover published data on the production of diphenolic acid from phenol and levulinic acid, two molecules that can be obtained from lignocellulosic biomass Although the production of DPA has been addressed in chapters of some previous reviews, it was necessary to prepare a specific work on this molecule and this is fulfilled with this contribution. The topic is adequately addressed, covering the most important aspects related to this chemical, but the following issues should contribute to improve the information provided in the manuscript: 

1. (DPA synthesis with homogeneous catalysis) Some information on the isolation of DPA from the reaction medium could help to obtain a clearer view of this homogeneous process and its drawbacks.

2. Tables should only include the different types of catalysts (for example, HCl, H2SO4, Amberlyst-15, among others) and the experimental conditions under which the best catalytic performance has been attained for each type of catalyst. Then, in the text, the influence of a particular experimental variable (such as the g(LA)/g(cat) ratio, the LA/phenol molar ratio, the reaction temperature or reaction time and the addition of additives) on the yield of p,p’-DPA can be explained. At the end of each section, conclusions could be drawn about the influence of these variables, which, together with the effect of the textural properties and the concentration/strength of the Brönsted acid sites of the catalysts (heterogeneous catalysis), would allow to develop new families of more active catalysts.

3. (Table 3, entries 15 y 16) How do the authors explain that using a high amount of Amberlyst-15 (4.4 versus 6.1 g(LA)/g(cat)), a much better yield of p,p’-DPA (55 versus 6) can be achieved at 60ºC instead of 120ºC, after 24 h?

4. (Related to point 2) Table 4 contains 33 entries, but most of the catalysts are based on Cs(x)H(3-x)PW12O40 and Cs(x)H(6-x)P2W18O62, so the recommendations given in point 2 could help to follow the information included in this Table.

5. The mechanism in the presence of Brönsted acid sites is clear, but what are the mechanistic pathways involved in the production of DPA in the presence of ionic liquids (section 2.5)?

6. An important issue related to heterogeneous catalysis is the reutilization of the catalyst, and this should deserve attention in this review.

7. Section 3 (applications of DPA: challenges and opportunities) offers relevant information to evaluate the potential of this molecule in different fields, which would contribute to having a more complete overwiew of this chemical.

8. Minor comments: (Table 2, entry 5) b should be as superscript; (Table 2, entry 10) instead . 

In summary, this review is very interesting and its publication could be recommended.

Author Response

This review deals with the synthesis and applications of diphenolic acid (DPA), considering that this molecule has emerged as a sustainable alternative for the replacement of bisphenol A with important toxicological problems. Review papers on a specific topic allow readers to have an updated vision of the progress made, and, in this sense, this paper tries to cover published data on the production of diphenolic acid from phenol and levulinic acid, two molecules that can be obtained from lignocellulosic biomass Although the production of DPA has been addressed in chapters of some previous reviews, it was necessary to prepare a specific work on this molecule and this is fulfilled with this contribution. The topic is adequately addressed, covering the most important aspects related to this chemical, but the following issues should contribute to improve the information provided in the manuscript: 

Reviewer: (DPA synthesis with homogeneous catalysis) Some information on the isolation of DPA from the reaction medium could help to obtain a clearer view of this homogeneous process and its drawbacks.

Authors: Regarding this observation of the Reviewer, the available literature on DPA recovery/purification is scarce (as already highlighted in different parts of the original manuscript). According to the already cited patent of Bader (reference [17] in the original submission; Bader, A.R. Resinous material. Granted Patent US2933472, 19 April 1960), the isolation of DPA from such a reaction environment can be realized by diluting the crude reaction mixture with water and next extraction with ethyl acetate. The extract (containing phenol and DPA) can be further extracted with aqueous solution of sodium bicarbonate, to selectively deprotonate the DPA, which becomes soluble in the water phase. The latter is acidified, extracted with ether and the corresponding extract is stripped in vacuo to yield the isolated DPA. Depending on the required purity degree, crude DPA can be further crystallized, generally from organic solvents, such as aromatic hydrocarbons (toluene), but also water or ethanol. According to the above purification procedure, the manuscript was modified, as follows:

“Isolation of DPA from the crude reaction mixture is a challenging topic, and only few examples are available in the literature, mainly applied to the DPA synthesized in the presence of a mineral acid (H2SO4 and HCl) as the catalyst. According to Bader [17], the isolation of DPA from such a reaction environment can be realized by diluting the crude reaction mixture with water and next extraction with ethyl acetate. The extract (containing phenol and DPA) can be further extracted with an aqueous solution of sodium bicarbonate, to selectively deprotonate the DPA, which becomes soluble in the water phase. The latter is acidified, extracted with ether and the corresponding extract is stripped in vacuo to yield the isolated DPA. Depending on the required purity degree, crude DPA can be further crystallized, generally from organic solvents, such as aromatic hydrocarbons (toluene), but also water or ethanol.”.

Reviewer: Tables should only include the different types of catalysts (for example, HCl, H2SO4, Amberlyst-15, among others) and the experimental conditions under which the best catalytic performance has been attained for each type of catalyst. Then, in the text, the influence of a particular experimental variable (such as the g(LA)/g(cat) ratio, the LA/phenol molar ratio, the reaction temperature or reaction time and the addition of additives) on the yield of p,p’-DPA can be explained. At the end of each section, conclusions could be drawn about the influence of these variables, which, together with the effect of the textural properties and the concentration/strength of the Brönsted acid sites of the catalysts (heterogeneous catalysis), would allow to develop new families of more active catalysts.

Authors: We thank the Reviewer for this observation. Anyway, we believe that the reported entries are necessary to better highlight and discuss the influence of the experimental variables (g(LA)/g(cat) ratio, LA/phenol molar ratio, reaction temperature, reaction time, presence and type of additive). However, according to the similar request 4 of the Reviewer, we have simplified the Table 4, keeping only the most valuable data of the cesium-based heteropolyacids (entries 56, 64, 65, 66 in the original version of the manuscript), removing the other ones (entries 57-63 in the original version of the manuscript), and data discussion was modified, accordingly. According to the Reviewer's suggestion, we have added a final critical sentence to each paragraph, to better highlight the corresponding key aspects useful for the development of new more efficient heterogeneous catalysts, as follows:

“Definitely, an overall evaluation of the data related to sulfonated catalysts highlights good catalytic performances for the Amberlyst-15 sulfonic resin and, even better, excellent performances for the synthesized Co@NC catalytic system, which is easy recoverable and thermally stable.” .

“In conclusion, considering the available data on heteropolyacids, the best catalytic performances are achieved with cesium-based systems, combining high yields/selectivity to the p,p’-DPA, with an excellent advantage on its recovery/reuse.”.

“On the basis of the above data, the use of zeolites for this condensation reaction is promising, considering that the zeolite pore size and structure must be adequate for the molecules of interest, as well as the type and strength of the acidity. In this regard, moderate strength of acid sites should be preferred for p,p’-DPA synthesis, in order to avoid undesirable side-reactions, typically occurring in the presence of strong Brønsted acid catalysts. For this purpose, Si/Al ratio represents the key parameter affecting the acid properties of these catalysts, determining not only the amount and concentration of acid sites but also their nature (Brønsted and Lewis) and strength. Therefore, Beta zeolite with a moderate aluminum content (H-Beta 19, Si/Al=23) represents the best catalyst to perform the solvent-free condensation between LA and phenol, owing to the shape selectivity conferred by its structure and to the adequate balance of acidity (Al content and speciation).”.

“According to the above data, the fine tunability of the catalytic properties of the ionic liquids certainly offers remarkable advantages for improving DPA synthesis, exploiting the Brønsted sulfonic groups to promote the LA conversion, according to the general mechanism already discussed for the other Brønsted acids (Figure 2). Moreover, the inclusion of a thiol group within the anion of the ionic liquid remarkably improved both yield and selectivity to p,p’-DPA. Although mechanicistic details have not been provided by the authors, the improved selectivity to the p,p'-DPA isomer could be attributed to the better cooperation between thiol group and cationic counterpart in the ionic liquid. However, the main bottleneck limiting the use of ionic liquids for such industrial application is their high cost, besides their uneasy isolation/reuse, requiring an additional separation step of the reaction mixture, by appropriate solvent extraction, given the high boiling points of the involved compounds.”.

Reviewer: (Table 3, entries 15 y 16) How do the authors explain that using a high amount of Amberlyst-15 (4.4 versus 6.1 g(LA)/g(cat)), a much better yield of p,p’-DPA (55 versus 6) can be achieved at 60ºC instead of 120ºC, after 24 h?

Authors: Regarding this observation of the Reviewer, the choice of a lower temperature (and a higher amount of Amberlyst-15) is effective for the production of the target p,p’-DPA isomer, whilst higher temperature (and a lower amount of Amberlyst-15) greatly promotes the formation of the unwanted o,p’-DPA one, as confirmed by the corresponding p,p’-DPA/o,p’-DPA molar ratios (entries 15 and 16 of Table 3). These results confirm that, working under similar conversion of levulinic acid, the temperature strongly influenced the selectivity of the reaction. On this basis, the following sentence has been added to the manuscript:

“Among the commercial heterogeneous sulfonic acid-based systems, Amberlyst-15 results the preferred choice, according to entries 15-17 of Table 3, the highest p,p’-DPA yield (55 mol%) and p,p’-DPA/o,p’-DPA molar ratio (15.8) were ascertained working at 60 °C for 24 h, employing a LA/phenol molar ratio of 1/4 and LA/catalyst weight ratio of 4.4. Remarkably, comparing the results obtained with the same LA/phenol ratio (1/4, according to entries 15 and 16), it is evident that a lower reaction temperature should be preferred, compensating with a greater amount of the catalyst, to get a similar conversion of LA, but a significantly better selectivity control towards the p,p’-DPA formation.

Reviewer: (Related to point 2) Table 4 contains 33 entries, but most of the catalysts are based on Cs(x)H(3-x)PW12O40 and Cs(x)H(6-x)P2W18O62, so the recommendations given in point 2 could help to follow the information included in this Table.

Authors: We thank the Reviewer for this comment. According to his/her suggestion, we have simplified the Table 4, keeping only the most valuable data of the cesium-based heteropolyacids (entries 56, 64, 65, 66 in the original version of the manuscript), and removing the other ones (entries 57-63 in the original version of the manuscript). On this basis, data discussion was accordingly modified.

Reviewer: The mechanism in the presence of Brönsted acid sites is clear, but what are the mechanistic pathways involved in the production of DPA in the presence of ionic liquids (section 2.5)?

Authors: The ionic liquids that showed to be active for the synthesis of DPA were those including hydrogen sulphate or sulphonic groups in their structures. On this basis, ionic liquids work alone as effective Brønsted acids, acting on the basis of the general reaction mechanism already discussed in the manuscript (see Figure 2 and the corresponding explanation, both already included in the original submission). Moreover, as for the previously discussed catalysts, also in this case, a synergistic positive effect is ascertained employing ionic liquids, with a remarkable improvement of both yield and selectivity to p,p’-DPA when the thiol group was included in the structure of the same ionic liquid, rather than being added apart as an additive. In this context, Liu et al. synthesised several ionic liquids having sulphonic groups and, for some of them, also thiol ones, the latter resulting of paramount relevance to obtain high p,p'-DPA/o,p’-DPA molar ratios. In fact, when a thiol, as mercaptoacetic acid, was employed as the additive, both an improvement of p,p'-DPA yield and a decrease in p,p'-DPA/o,p’-DPA molar ratio were observed, because thiol promoted the levulinic acid conversion but also the formation of the o,p’-DPA isomer. On the other hand, when the ionic liquids having thiol-containing anion were used, both levulinic acid conversion and p,p'-DPA/o,p’-DPA molar ratio were improved, showing that it had a key role for improving the yield and selectivity to p,p’-DPA. Although mechanicistic details have not been provided by the authors, the improved selectivity to the p,p'-DPA isomer can be attributed to the better synergy between thiol groups and cationic counterparts in the ionic liquid.

On this basis, the following sentences have been added in the manuscript:

“According to the above data, the fine tunability of the catalytic properties of the ionic liquids certainly offers remarkable advantages for improving the DPA synthesis, exploiting the Brønsted sulfonic groups to promote the LA conversion, according to the general mechanism already discussed for the other Brønsted acids (Figure 2). Moreover, the inclusion of a thiol group within the anion of the ionic liquid remarkably improved both yield and selectivity to p,p’-DPA. Although mechanicistic details have not been provided by the authors, the improved selectivity to the p,p'-DPA isomer could be attributed to the better cooperation between thiol groups and cationic counterparts in the ionic liquid. However, the main bottleneck limiting the use of ionic liquids for such industrial application is their high cost, besides their uneasy isolation, requiring an additional separation step of the reaction mixture, by appropriate solvent extraction, given the high boiling points of the in-volved compounds.”. 

Reviewer: An important issue related to heterogeneous catalysis is the reutilization of the catalyst, and this should deserve attention in this review.

Authors: The recyclability of the heterogeneous catalysts is surely an important issue, in particular in the perspective of an industrial development of this process. In the original submission, we have considered the recyclability issue of the proposed catalysts, when explicitly discussed by the authors. In particular, in some works, a washing of the spent catalyst is often proposed, anyway observing deactivation phenomena, in some cases already after the second catalytic run. The available data confirm that the main cause of the catalyst deactivation is the adsorption of organic matter on the catalyst surface, rather than the occurrence of leaching phenomena. On this basis, the calcination of the catalyst at temperatures higher than 400 °C is generally effective to restore the pristine performances. However, the calcination is feasible only for inorganic systems (sulfonated N-doped carbon nanotube, H3PW12O40-silica materials and zeolites), but not for the organic sulfonic resins (such as Amberlyst and Nafion), which are therefore less interesting for a desirable industrial scale-up. Anyway, such findings were already discussed in the corresponding paragraphs of the original submission, so we have better highlighted such key points in the Conclusions Section, as follows:

“Moreover, as discussed in detail, new ad hoc catalytic systems have been proposed by several authors, in some cases maintaining good catalytic performances after the reactivation step. In fact, catalyst deactivation was generally observed after its recycling, mainly due to the adsorption of organic matter on the catalyst surface. On this basis, the thermal calcination of the spent catalyst is often effective to restore its pristine performances. However, this thermal treatment is feasible only for inorganic systems (sulfonated N-doped carbon nanotube, H3PW12O40-silica materials and zeolites), whilst it is inappropriate for sulfonic acid-based organic resins (Amberlyst and Nafion), which would degrade at such high temperatures. Therefore, the inorganic catalysts are industrially more attractive, in the perspective of a desirable scale-up of the reaction already in the immediate future”.

Reviewer: Section 3 (applications of DPA: challenges and opportunities) offers relevant information to evaluate the potential of this molecule in different fields, which would contribute to having a more complete overview of this chemical.

Authors: We thank the reviewer for this positive comment.

 Reviewer: Minor comments: (Table 2, entry 5) b should be as superscript; (Table 2, entry 10) <NH2SO3H> instead <NH3SO3H>. 

Authors: We are sorry for these typo errors, and we have corrected those highlighted by the reviewer and other ones find in the manuscript.

Reviewer: In summary, this review is very interesting and its publication could be recommended.

Authors: Thank you again for your appreciation.